# Timing of *Smarcb1* and *Nf2* inactivation determines schwannoma versus rhabdoid tumor development

Jeremie Vitte [1], Fuying Gao[2], Giovanni Coppola [2], Alexander R. Judkins[3] & Marco Giovannini [1]

Germline mutations of the *SMARCB1* gene predispose to two distinct tumor syndromes: rhabdoid tumor predisposition syndrome, with malignant pediatric tumors mostly developing in brain and kidney, and familial schwannomatosis, with adulthood benign tumors involving cranial and peripheral nerves. The mechanisms by which *SMARCB1* germline mutations predispose to rhabdoid tumors versus schwannomas are still unknown. Here, to understand the origin of these two types of *SMARCB1*-associated tumors, we generated different tissue- and developmental stage-specific conditional knockout mice carrying *Smarcb1* and/or *Nf2* deletion. *Smarcb1* loss in early neural crest was necessary to initiate tumorigenesis in the cranial nerves and meninges with typical histological features and molecular profiles of human rhabdoid tumors. By inducing *Smarcb1* loss at later developmental stage in the Schwann cell lineage, in addition to biallelic *Nf2* gene inactivation, we generated the first mouse model developing schwannomas with the same underlying gene mutations found in schwannomatosis patients.

[1] Department of Head and Neck Surgery, David Geffen School of Medicine at UCLA and Jonsson Comprehensive Cancer Center (JCCC), University of California Los Angeles, Los Angeles, CA 90095, USA. [2] Semel Institute for Neuroscience & Human Behavior and Department of Psychiatry and Biobehavioral Sciences, University of California Los Angeles, Los Angeles, CA 90095, USA. [3] Department of Pathology and Laboratory Medicine, Children's Hospital Los Angeles, Keck School of Medicine, University of Southern California, Los Angeles, CA 90027, USA. Correspondence and requests for materials should be addressed to M.G. (email: mgiovannini@mednet.ucla.edu)

Germline alterations of *SMARCB1* gene predispose to two different inherited tumor syndromes: rhabdoid tumor predisposition syndrome (MIM 609322)[1] and familial schwannomatosis (MIM 162091)[2]. The first genetic evidence of the role of *SMARCB1* as a tumor suppressor was the identification of its biallelic mutations as the cause of most cases of malignant rhabdoid tumors (RTs)[3, 4], a highly aggressive pediatric cancer that usually occurs in the brain (named atypical teratoid rhabdoid tumor: AT/RT), kidneys and soft tissues in the first years of life. Heterozygous *SMARCB1* mutations are the basis of the rhabdoid tumor predisposition syndrome[3, 5]. More recently, *SMARCB1* has been identified as a predisposing gene in familial schwannomatosis[6], a condition characterized by the onset of multiple spinal, peripheral, and cranial-nerve schwannomas during adulthood in the absence of vestibular schwannomas[7]. Five percent of schwannomatosis patients also develop cranial or spinal meningiomas[8]. *SMARCB1* germline mutations have been found in 45% of familial probands and 7% of sporadic schwannomatosis patients[9]. Although the exact molecular pathogenetic mechanisms in these schwannomas remain to be elucidated, a 4-hit/3-step mechanism involving *SMARCB1* and *NF2* genes seems to underlie development of these benign tumors in schwannomatosis patients[10]. Germline mutations in *LZTR1*, a gene closely linked to *NF2* and *SMARCB1* on chromosome 22, have recently been identified in a significant proportion of schwannomatosis patients lacking *SMARCB1* germline mutations. Similar to *SMARCB1*-related schwannomas, different additional somatic mutations in *NF2* were identified in schwannomas from these patients, thus supporting the 4-hit/3-step hypothesis[11]. This mechanism reinstates the crucial role of biallelic *NF2* loss in schwannoma genesis and of developmental risk periods for *SMARCB1* and *NF2* mutations to occur. In contrast, a single case of double *SMARCB1/NF2* inactivation was found in RTs[12]. Recent analysis of a larger series confirmed *SMARCB1* as the primary tumor suppressor gene involved in the development of rhabdoid tumors with no recurrent additional oncogenic canonical pathway mutations identified[13]. This raises challenging questions about the molecular mechanisms by which germline mutations in the same gene predispose to early aggressive RTs versus late-onset benign PNS tumors. Analysis of the *SMARCB1* gene mutation spectrum points to genotype-phenotype correlations, with germline rhabdoid tumor mutations being more centrally placed in the coding sequence, involving multiple exons and truncating mutations of the *SMARCB1* gene. Conversely, schwannomatosis mutations are mostly non truncating mutations and located in hot spots at both 5′ and 3′ end of the *SMARCB1* gene[14]. However, families with RTs or multiple epitheloid schwannomas sharing the same *SMARCB1* mutation have also been described[15–17]. A link between schwannoma and RT has also been suggested by the histological analysis of a series of aggressive PNS tumors revealing rhabdoid features[18–20], and sporadic case reports of RTs emanating from cranial nerves[21–23].

Altogether, these observations raise the question of whether the two types of *SMARCB1*-deficient tumors arise from common or different progenitor cells. In mice, the specific deletion of the *Nf2* gene in P0 permissive cells targets the cells of origin of schwannomas in cranial nerves, peripheral nerves, and nerve roots[24, 25]. Although different hypotheses have been suggested, the cell of origin of RTs remains unclear. Different studies using histological and molecular markers or mouse models suggested that RTs could arise from the mesenchymal lineage[26], neural progenitor cells[27], neural crest stem cells[28–31], stem cells[32], germ and/or embryonic stem cells[33]. The homozygous inactivation of *Smarcb1* in mice leads to peri-implantation lethality and heterozygous mice develop RTs at low penetrance (15–30%)[29, 31, 34].

In two different *Smarcb1*[+/−] mouse models, RTs developed predominantly in the soft tissues of the head and neck[29, 34]. In another *Smarcb1*[+/−] model, 30% of the mice developed intra-cranial tumors and 27% in the spinal cord around the dorsal ganglia or spinal nerves that were classified as undifferentiated sarcomas with variable rhabdoid features[31]. All three types of *Smarcb1*[+/−] mice presented an extended time window for tumor onset with the earlier appearance at 3–4 months of age and median onset at 11–12 months of age, depending on the model[29, 31, 34]. Conditional *Smarcb1* inactivation in all tissues except the brain led to 100% of mice developing T-cell lymphoma with 13% of mice developing RTs with a median latency of 11 weeks after induction, demonstrating that inactivation of the second *Smarcb1* allele is rate limiting RT development in *Smarcb1*[+/−] mice[35].

To investigate the role of *Smarcb1* loss in PNS tumorigenesis, we conditionally inactivated *Smarcb1* in neural crest (NC) and Schwann cell (SC) lineages. We report that tumors arising from neural crest cells (NCCs) have histological and molecular characteristics of human RTs, providing novel clues to their cellular origin. Finally, we show that biallelic inactivation of *Smarcb1* and *Nf2* in SCs results in benign schwannoma, and not in RT, thus re-emphasizing the necessary and sufficient role of *NF2* loss in SC tumorigenesis, in both neurofibromatosis type 2 (NF2) and schwannomatosis, and the existence of a critical developmental risk period for *SMARCB1*-deficient RTs to occur.

## Results

**Smarcb1 loss in early NC promotes RT development**. We tested the hypothesis that *Smarcb1* inactivation in NCCs and in the SC lineage is sufficient to initiate schwannoma development in the mouse. Fifty percent of mice (3/6) carrying a germline *Smarcb1* deleted allele (*Smarcb1*[del/+]) developed tumors with rhabdoid features occurring over a wide age range (2–17 months of age). This observation is in line with the tumor phenotype described in other mouse models of *Smarcb1* inactivation[29, 31, 34]. Tumors were located in the brain (emergence of trigeminal nerve), spinal nerve roots and on the cheek and did not present histological features of schwannoma. To circumvent the early lethality of *Smarcb1*[−/−] mutants and malignant tumors found in *Smarcb1*[del/+] mice, we used a conditional floxed *Smarcb1* allele (*Smarcb1*[flox])[36]. To define the temporal window of susceptibility to *Smarcb1* loss, we used different promoters to induce Cre-mediated *Smarcb1* deletion at different stages of SC development. To specifically direct *Cre* recombinase expression to the NCCs at E9.5 and SC lineage we chose the promoter of the protein zero (P0) gene, which efficiently targets schwannoma precursor cells[24, 25]. *P0-CreC;Smarcb1*[flox/flox] mice, harboring a homozygous deletion of *Smarcb1*, were viable and obtained at the expected Mendelian ratio (21.6 vs. 25%, non-statistically significant difference, Fig. 1a).

*P0-CreC;Smarcb1*[flox/flox] mice presented with mild craniofacial abnormalities: orbit malformations and misshapen mandibles, with dental malocclusion eventually leading to outgrowing teeth. Few mice displayed bloated intestines with histological analysis revealing emaciated intestine wall associated with a reduced number of villi. About 65% of *P0-CreC;Smarcb1*[flox/flox] mice developed tumors between 1.5 and 5 months of age with an overall median survival of 3.2 months (Fig. 1b). These mice developed aggressive cranial nerve tumors: olfactory (91%, Fig. 1c, f, g), trigeminal (51%, Fig. 1i), oculomotor and optic (37%, Fig. 1m), or vestibulocochlear (14%, Fig. 1k, l) nerves were the most affected. Tumors were also found in the meninges with variable extent of brain invasion (14%, Fig. 2a, b), in spinal nerve roots (3%, Fig. 1n) and occasionally in the eye (Fig. 1m),

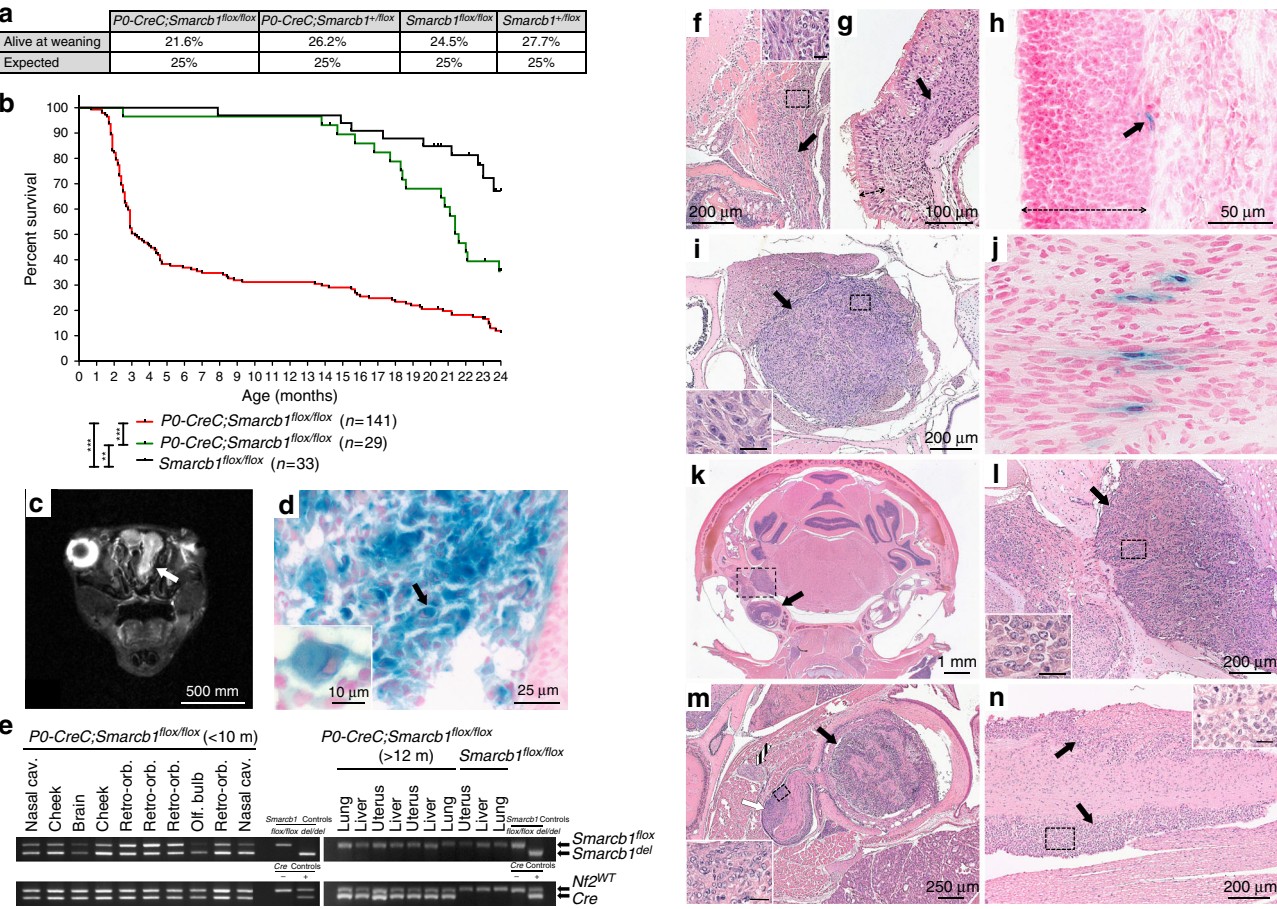

**Fig. 1** *P0-CreC;Smarcb1^flox/flox* mice develop RTs in NC-derived tissues. **a** Genotype distribution of mice at 3 weeks of age (*n* = 282) born from crossings of *P0-CreC;Smarcb1^flox/+* and *Smarcb1^flox/flox* mice. **b** Kaplan–Meier curve representing the percent survival of *P0-CreC;Smarcb1^flox/flox* (*n* = 141, median survival = 3.2 months), *P0-CreC;Smarcb1^flox/+* (*n* = 29, median survival = 21.6 months) and *Smarcb1^flox/flox* (*n* = 33, median survival > 24 months) mice versus age in months. **c** T2-weighted MR image identifying a tumor mass in the nasal cavities of a *P0-CreC;Smarcb1^flox/flox* mouse (*arrow*). **d** X-gal staining of a tumor from *P0-CreC;Smarcb1^flox/flox;ACZL* mice demonstrating activation of the beta-galactosidase reporter gene *ACZL* (*blue* staining) thus attesting for *Smarcb1* deletion throughout the tumor and in rhabdoid cells (*arrow and inset*). **e** *Smarcb1^del* allele is detected in all the tumors from young *P0-CreC; Smarcb1^flox/flox* mice (< 10 months of age). Long-term *P0-CreC;Smarcb1^flox/flox* survivor mice (Supplementary Fig. 2c) develop liver, lung, and uterus tumors, with no deletion of *Smarcb1* similar to control littermates (*Smarcb1^flox/flox*). These tumors are part of the normal tumor spectrum of aging FVB/N mice[24]. **f, g** *P0-CreC;Smarcb1^flox/flox* mice develop tumors along the olfactory nerve (*arrow*), close to the olfactory bulb (**f**) or in the nasal cavity below the olfactory epithelium (*double arrow*, **g**). **h** X-gal staining of the nasal cavity of a *P0-CreC;AZCL* mouse showing specific *Cre* recombinase expression in SCs of the olfactory nerve (*arrow*) below the olfactory epithelium layer (*double arrow*). **i** Tumor of the trigeminal nerve (transversal section). **j** X-gal staining of the trigeminal nerve of a *P0-CreC;AZCL* mouse showing specific *Cre* recombinase expression in SCs (*arrows*). **k** Tumor of the vestibulocochlear nerve (*dashed square*) invading the cochlea (*arrow*). Note that the contralateral side is not affected. **l** Detail of the vestibulocochlear nerve tumor shown in **k**. **m** Tumors in the eye (*black arrow*), optic nerve (*white arrow*), and oculomotor nerve (*striped arrow*) of a *P0-CreC;Smarcb1^flox/flox* mouse. **n** Tumors in the spinal nerve roots (*arrows*, longitudinal section). *Insets* in **f**, **i**, **l**, **m** and **n** represent a higher magnification of the *dashed square area* (*insets scale bar is* 20 μm). **\*\***(*P* < 0.01) and **\*\*\***(*P* < 0.001); log-rank (Mantel–Cox) test

with 62.5% (15/24) of the mice displaying multiple tumors at dissection. In all mice presenting with a head tilt and/or spinning behavior, unilateral tumors of the vestibular nerve ganglion and/or invasion of the inner ear were found (Fig. 1k, l). Use of the *P0-CreB* transgenic mice, which express Cre recombinase at E9.5, similarly to *P0-CreC*, but in a larger number of cells[25], resulted in a more severe phenotype with a lower percentage of viable *P0-CreB;Smarcb1^flox/flox* mice (1.4 vs. 25% expected, *P* < 0.0001, Supplementary Fig. 1a). Nasal cavity or retro-orbital tumors, with the same histological features than *P0-CreC;Smarcb1^flox/flox* tumors were found in 62.5% of *P0-CreB;Smarcb1^flox/flox* mice between 1.2 and 2 months of age (Supplementary Fig. 1b). Tumor location in *P0-CreC;Smarcb1^flox/flox* mice correlated with the expression pattern of the *P0-CreC* transgene (Supplementary Fig. 2a). X-gal staining of olfactory epithelium (Fig. 1h) and

trigeminal nerve (Fig. 1j) in *P0-CreC;ACZL* mice[25] demonstrated the presence of Cre activity in few cells that are likely the cells of origin of tumors in *P0-CreC;Smarcb1^flox/flox* mice. Tumors developed by *P0-CreC;Smarcb1^flox/flox;ACZL* mice before 10 months of age displayed a blue X-gal staining, demonstrating the Cre recombinase activity and *Smarcb1* deletion, including in rhabdoid cells (Fig. 1d). The presence of the *Smarcb1* deleted allele in all tumors from symptomatic *P0-CreC;Smarcb1^flox/flox* mice was confirmed by PCR analysis (Fig. 1e). Despite expression of the *Cre* recombinase and detection of the *Smarcb1* deleted allele, tumors were not found in kidneys, gonads and in brachial, saphenous and sciatic nerves of *P0-CreC;Smarcb1^flox/+* and *P0-CreC;Smarcb1^flox/flox* mice (Supplementary Fig. 2a, b). This finding can be explained either by the fact that *Cre* in *P0-CreC* and *P0-CreB* mice is not expressed in the cell of origin of kidney

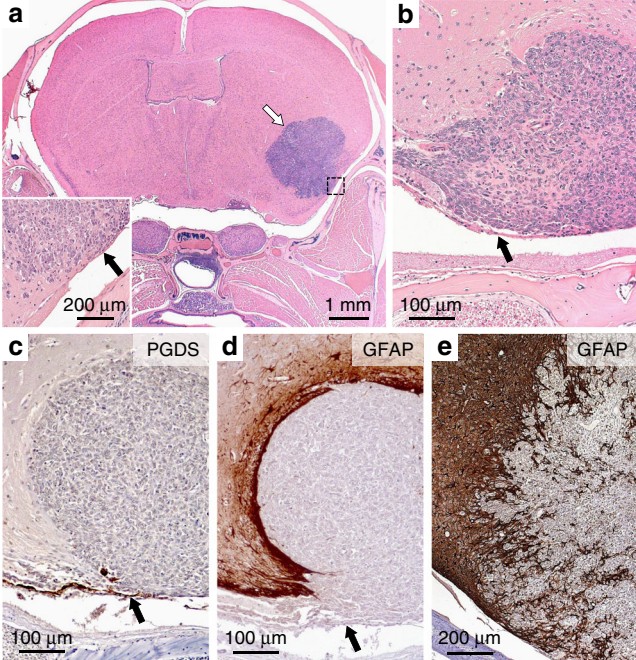

**Fig. 2** *P0-CreC;Smarcb1^flox/flox* mice develop RTs in the meninges. **a** Supratentorial tumors (*white arrow*) were found predominantly located in the subarachnoid space (*black arrow, inset*) with extension into the brain parenchyma. *Inset* represents a higher magnification of the *dashed square area.* **b** Detail of a tumor originating from the meninges. The *arrow* shows the arachnoid cell layer. **c** No PGDS staining was found in the tumor as opposed to the arachnoid cell layer (*arrow*). **d, e** The GFAP staining demonstrated the different degree (**d**, non-invasive; **e**, irregular tumor brain interface consistent with infiltration) of invasiveness of the tumors in the brain tissue

RTs, or is not expressed during the specific temporal window of vulnerability to RT development. No pathological alteration was found in nerves (cranial, phrenic, saphenous, sciatic nerves, and brachial plexus), DRGs and brain in long-term *P0-CreC; Smarcb1^flox/flox* survivor mice (up to 24 months of age).

**P0-CreC;Smarcb1^flox/flox tumors display human RT features.** Tumors of *P0-CreC;Smarcb1^flox/flox* mice displayed a histologic spectrum ranging from a primitive neuroectodermal (Fig. 3a), to a mesenchymal (Fig. 3b) or rhabdoid pattern (Fig. 3c). No epithelial differentiation was observed. Most tumor cells exhibited atypical large nuclei with vesicular chromatin, irregular and thick membrane, and prominent nucleoli (Fig. 3b *inset*). Some of these cells showed an eccentric nucleus associated with a prominent eosinophilic cytoplasmic inclusion (Fig. 3c and *inset*) or cytoplasmic vacuolar degeneration (Fig. 3d and *inset*), typical histological features encountered in human AT/RTs[37]. Rhabdoid features or scattered rhabdoid cells were found in all analyzed tumors, and in 30% of these, nests of rhabdoid cells were present (Fig. 3c). Bi-nucleated and giant multinucleated cells were also found in some tumors. Tumors displayed different patterns depending on their size at the time of dissection. Larger tumors typically presented more rhabdoid features than smaller tumors. All the tumors examined demonstrated loss of SMARCB1 nuclear expression by immunohistochemical staining (Fig. 3e). However, similar to human RTs[38], infiltrating inflammatory and endothelial cells retained SMARCB1 staining (Fig. 3e). The tumors behaved in an aggressive manner, displaying rapid growth, moderate to high cellularity and abundant mitotic figures

with high percentages of Ki-67-positive cells (50.7 ± 3.7% in neuroectodermal and 33.4 ± 3.9% in mesenchymal histological subtypes) (Fig. 3f). Larger tumors frequently showed areas of necrosis. Some tumors were contained within the boundaries of the tissue of origin, such as the epineurium of the trigeminal nerve (Fig. 1i), while other tumors, such as those extending in the cochlea from the vestibulocochlear nerve, were more invasive, (Fig. 1k, l). Supratentorial rhabdoid tumors were found predominantly located in the subarachnoid space with extension into the brain parenchyma (Fig. 2a, b). These tumors displayed well-demarcated smooth borders with adjacent tissues and reactive gliosis confirmed with GFAP staining in adjacent tissues to the tumor (Fig. 2d). In some cases, individual tumor cells were invading the brain parenchyma (Fig. 2e). The location of these tumors suggested that they originated outside of the brain parenchyma and developed from the cranial NC-derived meningeal progenitor cells[39]. The absence of PGDS staining (Fig. 2c), an arachnoid cell marker expressed postnatally in the NC-derived meninges, substantiates the hypothesis that meningeal RTs originate from early NC-derived progenitor cells, before differentiation in PGDS-positive arachnoid cells[39].

Because of their heterogeneous histologic features, RTs usually show a broad spectrum of immunohistochemical reactivity[40]. With a remarkable similarity to human RTs[37], tumors from *P0-CreC;Smarcb1^flox/flox* mice typically showed a consistent expression of vimentin (Fig. 3g), a moderate to strong SMA staining (Fig. 3h), whereas staining for cytokeratin proteins or neurofilament triplet proteins, commonly found in foci of human AT/RTs, was not detected (Fig. 3i, j). Staining of S100 protein, a marker of schwannoma[41], allowed to clearly distinguish the *P0-CreC;Smarcb1^flox/flox* mouse tumors from schwannomas: unlike schwannomas from *P0-CreC;Nf2^flox/flox* mice, which strongly and diffusely express S100 protein (Fig. 3k), tumor cells from *P0-CreC;Smarcb1^flox/flox* mice showed no S100 protein by IHC (Fig. 3l). Modest GFAP expression was present in *P0-CreC; Nf2^flox/flox* tumors (Fig. 3m) whereas *P0-CreC;Smarcb1^flox/flox* tumors showed little GFAP expression in few reactive astrocytes (Fig. 3n). However, tumors from both *P0-CreC;Smarcb1^flox/flox* and *P0-CreC;Nf2^flox/flox* mice displayed a strong staining for FABP7 (Fatty Acid Binding Protein 7) (Fig. 3o, p), an early marker of SC precursors[42]. Altogether, *P0-CreC;Smarcb1^flox/flox* mouse tumors displayed typical histological and immunohistochemical features of human RTs, clearly distinguishable from Nf2-deficient *P0-CreC;Nf2^flox/flox* schwannomas (Fig. 3 and Supplementary Table 1)[37].

**Molecular profiling classifies mouse RTs in three subgroups.** To address the relevance of the *P0-CreC;Smarcb1^flox/flox* mouse tumors as models of the human condition, we profiled 12 representative RTs by whole-genome RNA sequencing. Unsupervised hierarchical clustering of the top 5000 most variable transcripts in *P0-CreC;Smarcb1^flox/flox* RTs identified three molecular subgroups (Supplementary Fig. 3a). Based on the maximum cophenetic correlation score (non-negative matrix factorization (NMF) method), the optimal number of clusters was estimated to be two or three (Supplementary Fig. 3b, c). Across species unsupervised cluster analysis of the top 5000 most variable transcripts in *P0-CreC;Smarcb1^flox/flox* and human RTs from Johann et al., showed that mouse tumors distributed among the three recently identified human AT/RT molecular subgroups[43], independent of their anatomical origin (Fig. 4a). Thus, tumors from *P0-CreC; Smarcb1^flox/flox* mice recapitulate the molecular diversity of human AT/RTs. According to their inclusion in different human AT/RT clusters, we attributed similar molecular subgroups to the *P0-CreC;Smarcb1^flox/flox* RTs (mRT-Myc, mRT-Shh or mRT-Tyr). In

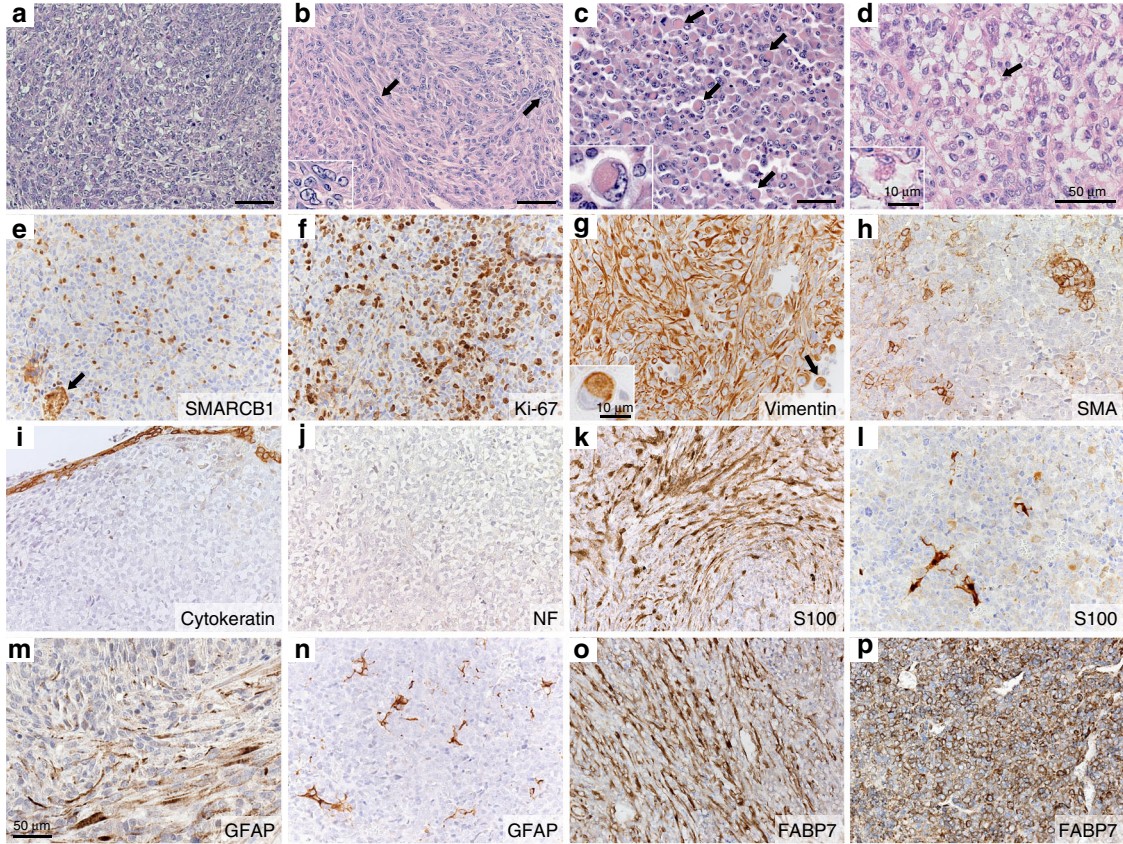

**Fig. 3** RTs of *P0-CreC;Smarcb1flox/flox* mice display typical histological and immunohistochemical staining features of human RTs. **a** Tumor presenting a primitive neuroectodermal pattern. **b** Tumor with mesenchymal pattern characterized by spindle-shaped cell component and nuclei showing vesicular chromatin staining pattern with prominent nucleoli (*arrows and inset*). **c** Nest of classic rhabdoid cells with eccentrically placed nuclei, vesicular chromatic staining and eosinophilic cytoplasm inclusion (*arrows and inset*). **d** Tumor demonstrating vacuolar cytoplasmic degeneration pattern (*arrow and inset*). **e** SMARCB1 staining shows loss of nuclear expression in tumor cells with retained expression in infiltrating inflammatory cells and intratumoral vasculature (*arrow*). **f** Ki-67 staining reveals high proliferative index. **g** Diffuse strong expression of vimentin in tumor tissue from *P0-CreC;Smarcb1flox/flox* mice. Note the staining of the cytoplasmic inclusions in rhabdoid cells (*arrow and inset*). **h** Foci of tumors cells show positive cytoplasmic staining for SMA. **i, j** No staining of cytokeratins **i** and neurofilament triplet proteins **j** were found in the tumors. **k** Strong S100 immunoreactivity in schwannoma from *P0-CreC;Nf2flox/flox* mice. **l** No S100 staining is present in the tumors cells of *P0-CreC;Smarcb1flox/flox* mice. Sparse positive cells are entrapped residual normal cells. **m** Moderate GFAP staining in *P0-CreC;Nf2flox/flox* schwannoma. **n** No GFAP staining is present in the tumors cells of *P0-CreC;Smarcb1flox/flox* mice. **o, p** Strong expression of FABP7, an early marker of SC precursors, in *P0-CreC;Nf2flox/flox* **o** and *P0-CreC;Smarcb1flox/flox* **p** tumors. *Scale bars*: 50 μm for main pictures and 10 μm for *insets*

a unsupervised cluster analysis of gene expression profiles from *P0-CreC;Smarcb1flox/flox*, *Rosa26-CreERT2;Smarcb1flox/flox*[30] and *hGFAP-Cre;Smarcb1flox/flox;Trp53flox/flox*[44] RT mouse models, tumors segregated based on molecular subgroups rather than on the model of origin, demonstrating their molecular analogy (Fig. 4b). We conducted Weighted Gene Co-expression Network Analysis (WGCNA), a systems biology approach used to identify networks of co-expressed genes in relation to phenotypic data, using the R package. WGCNA allowed identification, within the highest positively correlated module with each molecular subgroup, of several representative marker genes that were significantly and consistently overexpressed in both human and mouse RT molecular subgroups (Fig. 4c). Specifically, *TRPM3* (Transient Receptor Potential cation channel subfamily M member 3), a membrane channel that can promote renal cell tumors[45], was found consistently overexpressed in the human hTYR and mouse mRT-Tyr molecular subgroups (Fig. 4c). *HDGFRP3* (Hepatoma-Derived Growth Factor, Related Protein 3), which plays an essential role in hepatocellular carcinoma pathogenesis[46] and *LOXL4* (Lysyl Oxidase Like 4), which promotes proliferation and metastasis of gastric cancer[47], were consistently overexpressed in mouse and human SHH and

MYC molecular subgroups, respectively (Fig. 4c). *TYR, MYCN,* and *HOTAIR*, marker genes of the three human subgroups[43] showed a similar trend of overexpression in the respective mouse RT subgroups, albeit not statistically significant likely due to the small number of mouse tumors analyzed (Fig. 4c).

As in human AT/RTs[48], *Smarcb1* deficiency in all *P0-CreC;Smarcb1flox/flox* RT subgroups was correlated with an increased expression of the polycomb gene *Ezh2* (Supplementary Fig. 4a, b), a key driver of oncogenesis[49]. Analysis of molecular pathways in *P0-CreC;Smarcb1flox/flox* RTs by western blot showed increased phosphorylation of Akt, S6, and 4E-BP1, consistent with activation of the mTOR/Akt pathway, similar to schwannomas in *P0-CreC;Nf2flox/flox* mice (Supplementary Fig. 4c). Cell cycle proteins (CDK4, CDK6, cyclin D1, p16) were overexpressed in all *P0-CreC;Smarcb1flox/flox* tumors and phosphorylation of the retinoblastoma protein was found in 3 out of 5 RTs (Supplementary Fig. 4c). Activation of these pathways has been reported in human RTs[50].

**Early NC molecular signature in *Smarcb1*-deficient RTs.** To explore the effect of *Smarcb1* deficiency on the SWI/SNF complex in *P0-CreC;Smarcb1flox/flox* mouse RTs, we analyzed the

expression of the different complex subunits using the RNA-Seq gene expression data. High expression of *Actl6a* (BAF53a), *Phf10* (BAF45a), *Dpf2* (BAF45d), and *Ss18* (SYT) genes, encoding subunits found in the pluripotent embryonic stem cell (esBAF) and in the multipotent neural progenitor (npBAF) complexes[51]

was found in RTs of all three molecular subgroups (Fig. 4d). In contrast, mouse RTs did not express *Actl6b* (BAF43b), *Dpf1* (BAF45b), *Dpf3* (BAF45c) and *Ss18l1* (CREST) genes, encoding subunits exclusively found in the postmitotic neural BAF complex (nBAF) (Fig. 4d). These results, except for *SS18L1* (CREST), were

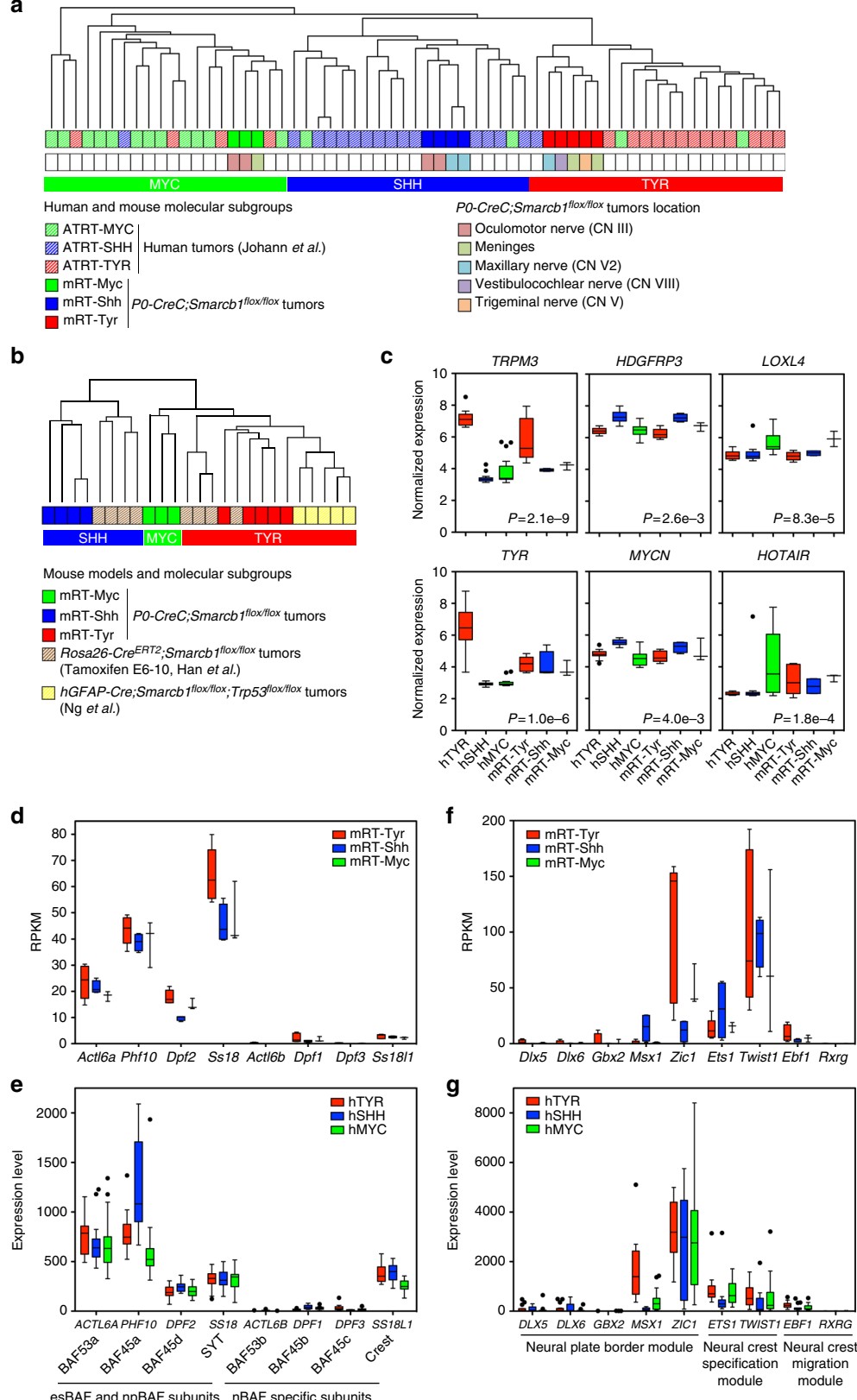

consistent with subunit gene expression in the series of human AT/RTs from Johann et al. (Fig. 4e) and demonstrated that the residual SWI/SNF complex in Smarcb1-deficient RTs is characteristic of undifferentiated progenitor cells[51], pointing to an early developmental cell population of origin for RTs. Analysis of gene regulatory networks underlying formation of NCCs showed robust expression of genes exclusively expressed in the neural plate border and NC specification modules[52] in both P0-CreC;Smarcb1[flox/flox] and human RT samples (Fig. 4f, g). The fact that human AT/RTs and RTs from P0-CreC;Smarcb1[flox/flox] mice retain specific marker genes of early NC formation, strongly corroborates the hypothesis of an early NCC as their cell of origin.

**SC differentiation suppresses Smarcb1-driven tumorigenicity.** The existence of adult carriers of a SMARCB1 mutation without RT in families presenting either RTs or schwannomas suggests the existence of a specific developmental time window during which RT progenitor cells are vulnerable to SMARCB1 loss[15–17]. To assess if loss of Smarcb1 at a later stage of development would induce tumorigenesis, we bred Smarcb1[flox] mice with DHH-Cre and mGFAP-Cre transgenic mice that express the Cre recombinase at later stages of SC development. DHH-Cre transgenic mice[53] express Cre in SC precursors beginning at embryonic day E12.5[42]. DHH-Cre;Smarcb1[flox/flox] mice had an overall median survival of 3 weeks and displayed progressive hindlimb paralysis. The sciatic nerves were thinner and more transparent than in control littermates. Histological analysis demonstrated that loss of Smarcb1 (Supplementary Fig. 5c, d) led to a lack of structure and organization of the nerve fibers (Supplementary Fig. 5a, b) with increased numbers of BrdU and Ki-67 positive cells (Supplementary Fig. 5e–h). No tumors were found in cranial nerves, peripheral nerves, or dorsal root ganglions (DRGs) of DHH-Cre;Smarcb1[flox/flox] mice during their short lifespan.

To target immature SCs, we generated mGFAP-Cre;Smarcb1[flox/flox] mice using the mGFAP-Cre transgenic mice that express the Cre recombinase under the promoter of the mouse GFAP gene[54]. This promoter targets immature SCs in the peripheral nervous system starting at E13.5[54], postnatal astrocytes throughout the CNS and postnatal astroglial cells in the cerebellum[55]. X-gal staining of tissues from mGFAP-Cre;ACZL mice demonstrated expression of Cre recombinase in spinal nerve root SCs (Supplementary Fig. 6d), spinal cord and throughout the brain (Supplementary Fig. 6c). At three months of age, mGFAP-Cre;Smarcb1[flox/flox] and mGFAP-Cre;Smarcb1[del/flox] mice started developing cataracts, displayed progressive ataxia with loss of motor coordination and gait balance, eventually leading to loss of bodyweight and euthanasia. This neurological phenotype was similar to the one described at earlier age in hGFAP-Cre;Smarcb1[flox/flox56] and hGFAP-Cre;Smarcb1[flox/flox];Trp53[flox/flox44] mice. The median survival of mGFAP-Cre;Smarcb1[flox/flox] and mGFAP-Cre;Smarcb1[del/flox] mice was 3.7 and 3.8 months of age (P = 0.74, non-statistically significant difference, Supplementary Fig. 6b), respectively. Histological analysis of the brain revealed atrophy of the cerebellum, but no tumors were found in the brain, cranial nerves or DRGs (Supplementary Fig. 6e, f). No change in cellularity was found in the DRGs of mGFAP-Cre;Smarcb1[flox/flox] mice compared with control littermates (Fig. 5a, b, g).

In conclusion, the observation that DHH-Cre;Smarcb1[flox/flox] and mGFAP-Cre;Smarcb1[flox/flox] mice did not develop RTs allowed us to narrow down to early NC development the time window for Smarcb1 loss to promote malignant tumorigenesis.

**Smarcb1 and Nf2 loss in SCs promotes schwannoma development.** The observation of frequent somatic, tumor-specific NF2 mutations, and the loss of the second NF2 allele in schwannomas from patients with germline SMARCB1 mutations[10, 57–59] strongly suggest that the classical two-hit model of tumorigenesis does not pertain in the tumors of schwannomatosis patients, as it would require biallelic SMARCB1 inactivation to be sufficient for tumor initiation or growth[7]. Thus, we tested the hypothesis that NF2 loss is necessary for schwannoma formation in schwannomatosis patients with germline SMARCB1 mutations.

To model the different steps leading to the double SMARCB1/NF2 gene loss, we generated combinatorial Smarcb1 and Nf2 wild-type or mutant alleles, representative of the 4-hit/3-step model. P0-CreC;Smarcb1[flox/flox];Nf2[flox/flox] mice were not viable, demonstrating that loss of both Smarcb1 and Nf2 during early development is lethal. However, in mGFAP-Cre;Smarcb1[flox/flox];Nf2[flox/flox] mice, combination of Smarcb1 and Nf2 inactivation resulted in viable mice, with a median survival of 6.4 months of age. The statistically significant longer survival is due to the fact that these mice do not develop the rapidly growing RTs inevitably causing death in P0-CreC;Smarcb1[flox/flox] mice. Cre is expressed in the DRGs of mGFAP-Cre mice starting at E13.5[54]. The mGFAP-Cre;Smarcb1[flox/flox];Nf2[flox/flox] mice developed tumorlets consisting of whorls of SCs in the DRGs (Fig. 5e). These DRG lesions are similar to those described in the P0-CreC;Nf2[flox/flox25] and mGFAP-Cre;Nf2[flox/flox] mice, and are reminiscent of the schwannoma tumorlets found in NF2 patients[60], thus demonstrating the necessary role of Nf2 loss for schwannoma formation. Schwannoma tumorlets in the DRGs of mGFAP-Cre;Smarcb1[flox/flox];Nf2[flox/flox] mice were negative for SMARCB1 and merlin staining, about 60% were positive for S100 protein staining, and all were positive for GFAP and FABP7 (Fig. 6). The mGFAP-Cre;Smarcb1[flox/flox];Nf2[flox/+] mice showed similar

**Fig. 4** Molecular profiling of RTs from P0-CreC;Smarcb1[flox/flox] mice. **a** Unsupervised hierarchical clustering of gene expression profiles from 12 P0-CreC;Smarcb1[flox/flox] mouse tumors and 49 human AT/RTs[43] using the 5,000 most variable transcripts. The P0-CreC;Smarcb1[flox/flox] tumors distributed among the three human AT/RT molecular subgroups. Information about tumor molecular subgroups and anatomical location is displayed in the lower bars. **b** Unsupervised hierarchical clustering of gene expression profiles from 12 P0-CreC;Smarcb1[flox/flox] mouse tumors, eight tumors from the tamoxifen inducible Rosa26-Cre[ERT2];Smarcb1[flox/flox] mouse model[30] and five tumors from the hGFAP-Cre;Smarcb1[flox/flox];Trp53[flox/flox] mouse model[44] using the 5,000 most differentially expressed genes. Information about tumor molecular subgroups is displayed in the lower bars. **c** Normalized expression level of representative genes for both of the three human AT/RT molecular subgroups: hTYR (n = 16), hSHH (n = 16), and hMYC (n = 17) and the three P0-CreC;Smarcb1[flox/flox] RT molecular subgroups: mRT-Tyr (n = 5), mRT-Shh (n = 4) and mRT-Myc (n = 3). On the basis of the cluster analysis in (**a**), we respectively named hMYC, hSHH and hTYR the three tumor subgroups including human tumors from the MYC, SHH and TYR molecular subgroups[43]. The P-value indicates the statistical significance of gene enrichment within their respective WGCNA module for each respective molecular subgroup. **d–g** Gene expression levels of specific SWI/SNF complex subunits (**d**, **e**) and NC developmental markers (**f**, **g**) in the three molecular subgroups of P0-CreC;Smarcb1[flox/flox] RTs (**d**, **f**) and human AT/RTs (**e**, **g**). **d**, **f** RNA-Seq data are expressed in RPKM (Reads Per Kilobase of transcript per Million mapped reads). **e**, **g** graphs of microarray gene-expression intensity from dataset GSE70678. **c–g** RNA-Seq and microarray gene expression levels are shown using Tukey boxplots: the central lines represent the median, boxes (interquartile range or IQR) represent 50% of data ranging from the 25 to 75% quantile, whiskers represent extremes up to 1.5-fold box size, circles show outliers

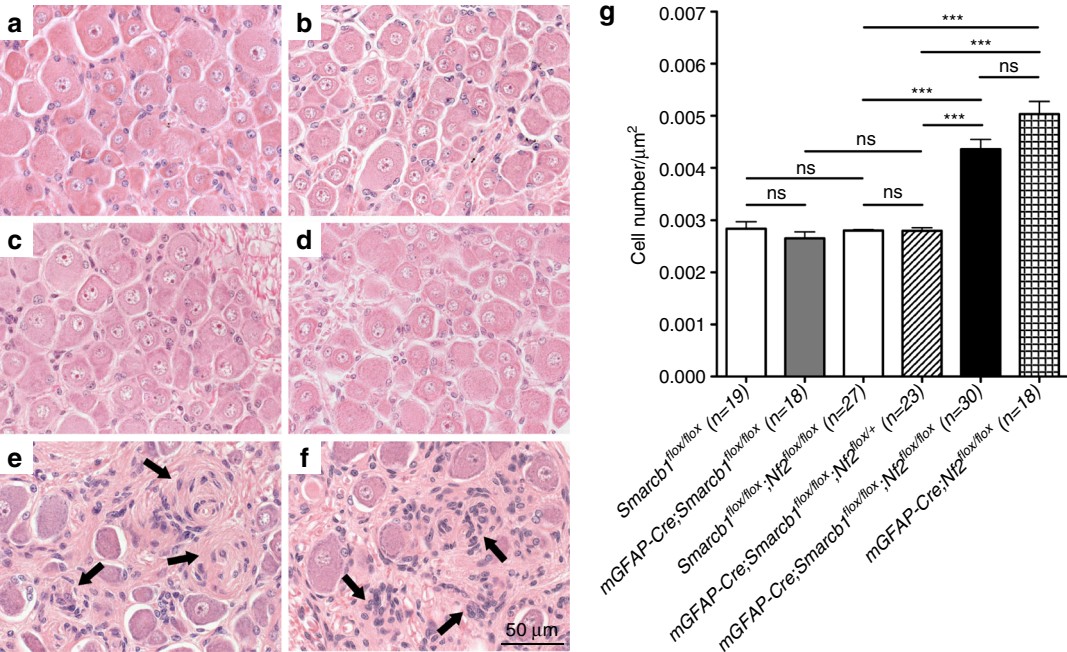

**Fig. 5** Biallelic loss of *Nf2* is necessary to induce schwannoma formation. **a**, **b** No difference in cellularity of DRGs from *mGFAP-Cre;Smarcb1$^{flox/flox}$* mice (**b**) compared with *Smarcb1$^{flox/flox}$* control mice (**a**). **c**–**f** The loss of one additional *Nf2* allele in *mGFAP-Cre;Smarcb1$^{flox/flox}$;Nf2$^{flox/+}$* mice does not impair the DRG cellularity (**d**) compared to *mGFAP-Cre;Smarcb1$^{flox/flox}$* mice (**b**) or *Smarcb1$^{flox/flox}$;Nf2$^{flox/flox}$* control mice (**c**). The biallelic loss *Nf2* induced the formation of schwannoma tumorlets in *mGFAP-Cre;Smarcb1$^{flox/flox}$;Nf2$^{flox/flox}$* mice (**e**), similar to *mGFAP-Cre;Nf2$^{flox/flox}$* mice (**f**). **g** Number of cells per DRG surface for each genotype. Data are represented as mean ± s.e.m.; *n* is the number of DRG analyzed per genotype, ns (not significant), ***(*P* < 0.001), two-tailed unpaired Student's *t*-test

phenotype to *mGFAP-Cre; Smarcb1$^{flox/flox}$;Nf2$^{flox/flox}$* mice (cataracts, impaired motor coordination and gait balance), leading to a median survival of 7.1 months of age. Similarly to *mGFAP-Cre;Smarcb1$^{flox/flox}$* mice, the decreased and altered mobility of *mGFAP-Cre;Smarcb1$^{flox/flox}$;Nf2$^{flox/flox}$* and *mGFAP-Cre;Smarcb1$^{flox/flox}$;Nf2$^{flox/+}$* mice led to bodyweight loss and euthanasia. Interestingly, no schwannoma tumorlets were found in the DRGs of *mGFAP-Cre;Smarcb1$^{flox/flox}$;Nf2$^{flox/+}$* mice, which contained similar cellularity as DRGs of *Smarcb1$^{flox/flox}$; Nf2$^{flox/flox}$* control mice (0.0028 vs. 0.0028 cells/μm$^2$, *P* = 0.94) (Fig. 5c, d, g) and *mGFAP-Cre;Smarcb1$^{flox/flox}$* mice (0.0028 vs. 0.0026 cells/μm$^2$, *P* = 0.35) (Fig. 5b, d, g). However, DRGs of *mGFAP-Cre;Smarcb1$^{flox/flox}$;Nf2$^{flox/flox}$* mice showed a significantly increased cellularity compared to those of *mGFAP-Cre; Smarcb1$^{flox/flox}$;Nf2$^{flox/+}$* mice (0.0044 vs. 0.0028 cells/μm$^2$, *P* = 0.0008) (Fig. 5d, e, g), thus demonstrating the crucial and necessary role of biallelic *Nf2* loss for schwannoma formation. Ki-67 staining identified the presence of few proliferating cells in the DRGs, including in the *mGFAP-Cre;Smarcb1$^{flox/flox}$; Nf2$^{flox/flox}$* and *mGFAP-Cre;Nf2$^{flox/flox}$* schwannoma tumorlets (Fig. 6). Surprisingly, complete loss of *Smarcb1* in the *mGFAP-Cre;Smarcb1$^{flox/flox}$;Nf2$^{flox/flox}$* tumors did not result in increased cell proliferation compared to schwannoma tumorlets in DRGs from *mGFAP-Cre;Nf2$^{flox/flox}$* mice (Fig. 5f, g). The very low percentage of proliferating cells in the DRGs of *mGFAP-Cre; Smarcb1$^{flox/flox}$;Nf2$^{flox/flox}$* mice reflects the benign nature of these tumors compared with malignant RTs in *P0-CreC;Smarcb1$^{flox/flox}$* mice (Figs. 3, 5 and 6).

## Discussion
Patients with germline mutations of the *SMARCB1* gene are predisposed to develop RTs or schwannomas, two tumor types with dramatically different clinical features. However,

the presence of aggressive PNS tumors with rhabdoid features suggests a link between the development of RTs and schwannomas[20]. We used the *P0-CreC* mouse line, known to express the Cre recombinase in NCCs, to target the precursors of the SC lineage and the schwannoma cells of origin[42] (Fig. 7). The early onset and high penetrance of RTs in *P0-CreC; Smarcb1$^{flox/flox}$* mice demonstrated that loss of *Smarcb1* in early NC (E9.5) is necessary for RT tumorigenesis (Fig. 7). This result is consistent with the high penetrance of AT/RTs in *Rosa26-CreERT2;Smarcb1$^{flox/flox}$* mice injected with tamoxifen between E6 and E10[30]. Although the *Rosa26-CreERT2* model doesn't target *Smarcb1* deletion in a specific cell type, expression profiling suggests that the cell of origin could be ectomesenchyme, a cephalic NCC-derived mesoderm or neural progenitors depending of the tumor subtype[30]. Inactivation of *Smarcb1* at later stages of SC development failed to initiate tumorigenesis in *DHH-Cre; Smarcb1$^{flox/flox}$* and *mGFAP-Cre;Smarcb1$^{flox/flox}$* mice suggesting that *Smarcb1* loss is not tumorigenic in the peripheral SC lineage or in the brain glial cells when the deletion is induced after E12.5 or E13.5, respectively (Fig. 7); or that the neurological deficits were lethal before the mice developed tumors. This result is consistent with other studies demonstrating that single loss of *Smarcb1* in *hGFAP*-positive neural precursors is not tumorigenic[44, 56] and that additional loss of *Trp53* is required to induce AT/RT development in *hGFAP-Cre;Smarcb1$^{flox/flox}$; Trp53$^{flox/flox}$* mice[44]. The use of different Cre drivers allowed us to narrow down the temporal window during which *Smarcb1* loss is tumorigenic. The existence of a specific developmental time window could also explain the presence of RTs and schwannomatosis within a family sharing the same *SMARCB1* mutation[15–17]. The neurological deficits developed by *DHH-CreC;Smarcb1$^{flox/flox}$*, *mGFAP-Cre;Smarcb1$^{flox/flox}$* and *mGFAP-Cre;Smarcb1$^{del/flox}$* mice with late *Smarcb1* inactivation also point to a role of *Smarcb1* in nervous system development, which could

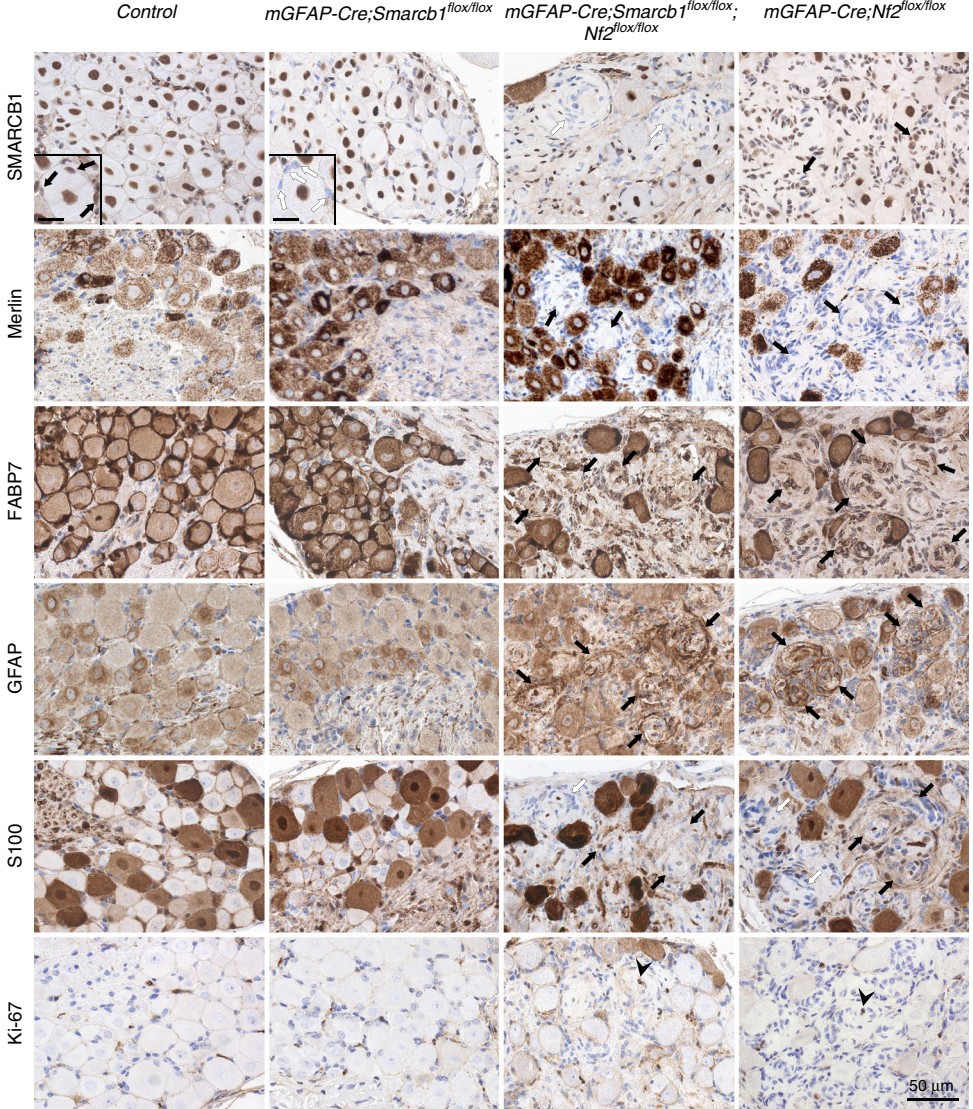

**Fig. 6** Histological characterization of DRGs and schwannoma tumorlets in control, *mGFAP-Cre;Smarcb1*[flox/flox], *mGFAP-Cre;Smarcb1*[flox/flox];*Nf2*[flox/flox], and *mGFAP-Cre;Nf2*[flox/flox] mice. DRGs from *mGFAP-Cre;Smarcb1*[flox/flox] mice present loss of SMARCB1 nuclear staining (*white arrows*), but no tumorlets or increased Ki-67 immunopositive staining. Tumorlets in DRGs of *mGFAP-Cre;Smarcb1*[flox/flox];*Nf2*[flox/flox] mice show loss of both SMARCB1 and merlin (*white and black arrows*), and tumorlets in DRGs of *mGFAP-Cre;Nf2*[flox/flox] mice show loss of merlin only (*black arrows*). Both tumorlets from *mGFAP-Cre; Smarcb1*[flox/flox];*Nf2*[flox/flox] and *mGFAP-Cre;Nf2*[flox/flox] mice show strong positive staining for FABP7 and GFAP (*black arrows*). Tumorlets from these both genotypes are S100 positive (*black arrows*) or negative (*white arrows*). Ki-67 positive staining is found in tumorlets of both *mGFAP-Cre;Smarcb1*[flox/flox]; *Nf2*[flox/flox] and *mGFAP-Cre;Nf2*[flox/flox] mice (*arrowhead*). *Inset scale bar*: 15 μm

underlie the neurological deficits found in the *SMARCB1*-related human Coffin–Siris syndrome[61].

The fact that most of the tumors were found in the head of *P0-CreC;Smarcb1*[flox/flox] mice is consistent with early *Cre* expression under the control of the P0 promoter in cranial NCCs and the prominent expression of *Smarcb1* between E8 and E11.5 during the embryonic development of the headfolds, neural folds and first brachial arch[34]. P0 was originally identified as a SC-specific myelin structural protein, but was more recently found expressed by migrating NCCs in the head of E10.5 mouse embryos[62], as well as in the otic placode and migrating cranial NCCs of E10 rat embryos. After E11, P0 can be detected in the ophthalmic and submandibular branches of the trigeminal nerves, within the trigeminal ganglion and in the developing acoustic/facial nerves of rat embryos. The *P0-CreC* transgene used in this study is expressed in the head of mouse embryos as early as E9.5[25]. The presence of craniofacial abnormalities confirms the early

inactivation of *Smarcb1* in the cranial NC, which is at the origin of multiple head structures such as bones and cartilages, SCs, olfactory ensheathing cells and meninges. The high frequency of tumors originating in the olfactory nerve is likely due to the fact that olfactory ensheathing cells, as opposed to the non-myelinating SCs of the trunk, do not down regulate P0 expression. Therefore, the promoter driving the *Cre* recombinase is continuously activated in the olfactory ensheathing cells of *P0-CreC;Smarcb1*[flox/flox] mice. Activity of the *P0-Cre* transgene has also been demonstrated in the enteric ganglia[63], thus explaining the presence of bloated intestine resulting from defects of the enteric nervous system, which is derived from vagal and sacral NCCs. The presence of RTs in the subarachnoid space, in the rostral part of the cortex is consistent with the hypothesis that these tumors developed from the cranial NC-derived meningeal progenitor cells[39]. Although in patients AT/RTs are predominantly infratentorial, they can also develop in the

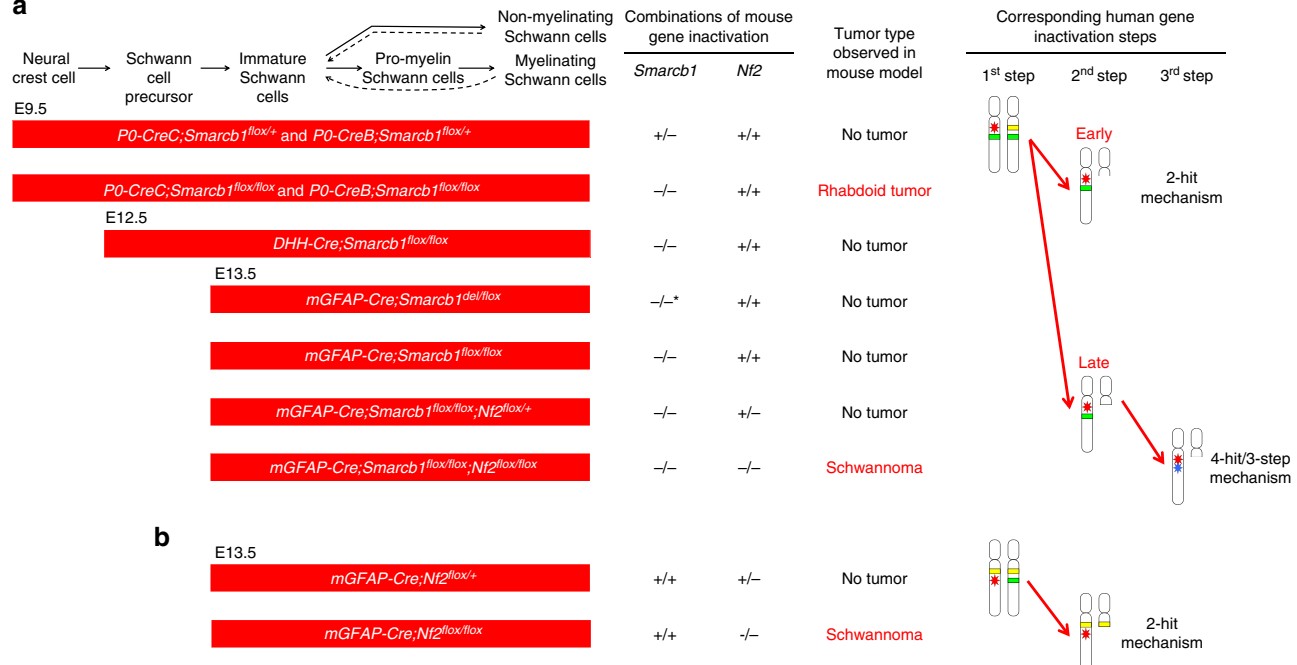

**Fig. 7** Proposed model of schwannoma versus RT initiation in the NC-derived SC lineage. Schematic illustration of the different combinations of mouse genotypes modeling different steps of the human 4-hit/3-step mechanism sequence and corresponding tumor phenotypes. **a** *Smarcb1* loss in early NC results in RT development in *P0-CreC;Smarcb1^{flox/flox}* mice, but is not tumorigenic when occurring at later SC developmental stages (*DHH-Cre;Smarcb1^{flox/flox}* and *mGFAP-Cre;Smarcb1^{flox/flox}* mice). Thus, loss of *Smarcb1* does not contribute to schwannoma development in *mGFAP-Cre;Smarcb1^{flox/flox};Nf2^{flox/flox}* mice where *Nf2* loss in the SC lineage is sufficient for tumor initiation, as **b** in *mGFAP-Cre;Nf2^{flox/flox}* mice. *germline *Smarcb1* allele inactivation

supratentorial region of the brain with frequent connection to the ventricles[64]. Human RTs have also been described in the meninges[65]. The early NC origin of *P0-CreC;Smarcb1^{flox/flox}* RTs is also supported by expression of early NC specification markers and SWI-SNF subunits characteristic of undifferentiated cells. Altogether, these results strongly support the hypothesis that both PNS and meningeal RTs originate from NCCs.

Tumors found in *P0-CreC;Smarcb1^{flox/flox}* mice displayed histological features strikingly similar to human RTs, including aggressive growth, characteristic histological patterns and presence of rhabdoid cells[40]. The anatomical locations of RTs in *P0-CreC;Smarcb1^{flox/flox}* mice have all been described in human RT patients: optic[22], oculomotor[66], trigeminal[21], vestibulochochlear[23] nerves, eye[67], nasal cavities[68], meninges[65] and spine[69]. About 60% of *P0-CreC;Smarcb1^{flox/flox}* mice displayed multiple tumors at the time of dissection, which is consistent with the occurrence of synchronous RTs in some patients with *SMARCB1* mutations[66]. Finally, molecular profiling showed that *P0-CreC;Smarcb1^{flox/flox}* RTs represented the full spectrum of human AT/RT molecular subtypes[43].

Although in *mGFAP-Cre;Smarcb1^{flox/flox};Nf2^{flox/flox}* mice loss of *Smarcb1* and *Nf2* is synchronous and not sequential as postulated by the 4-hit/3-step model, this is the first tumor model reproducing the genetic profile of schwannomatosis-schwannomas with concomitant loss of both *Smarcb1* and *Nf2* genes, with schwannoma and no RT development (Fig. 7). The *mGFAP-Cre;Smarcb1^{flox/flox};Nf2^{flox/flox}* mouse model demonstrates that biallelic loss of *Smarcb1* loss is dispensable and *Nf2* loss is necessary for schwannoma development[41], as in *mGFAP-Cre;Nf2^{flox/flox}* mice. This is consistent with the presence of *NF2* somatic mutations in the vast majority of schwannomas in schwannomatosis patients[10, 11]. Similar to schwannomatosis patients, loss of both *Smarcb1* and *Nf2* does not increase the malignancy of *mGFAP-Cre;Smarcb1^{flox/flox};Nf2^{flox/flox}* tumors

compared to *mGFAP-Cre;Nf2^{flox/flox}* tumors with single *Nf2* loss. In contrast to NF2, schwannomas in schwannomatosis patients are typically distinctly painful, rather than manifesting with localized neurologic deficits[2]. Thus, *Smarcb1* inactivation could be responsible for the pain phenotype, as suggested by a report showing that loss of *Smarcb1* in adult mouse SC induces the expression of a secreted factor that in turn increases TRPV1 expression and thermal sensitivity[70].

In conclusion, we found that biallelic inactivation of *Smarcb1* in the NC leads to the development of PNS and meningeal RTs, thus identifying the cell of origin for these RT subsets in an early NC population. We also defined an early spatio-temporal window during which *Smarcb1* loss results in malignant tumor formation and showed that biallelic *Nf2* loss is necessary and *Smarcb1* loss is dispensable for schwannoma formation. This mouse model developing schwannomas with the same underlying gene mutations found in schwannomatosis patients will prove invaluable for the study of other schwannomatosis-associated phenotypes, such as neuropathic pain.

## Methods

**Mice and genotyping**. All mouse strains were maintained on FVB/N genetic background. Mice were monitored twice a week for 24 months or until a tumor or evidence of a tumor (paralysis, swelling, lethargy) was observed. All animal care and experimentation were performed with the approval of the Institutional Animal Care and Use Committees under protocol number HE1175-10-02. *Smarcb1^{del/+}* mice were obtained by breeding *mGFAP-Cre;Smarcb1^{flox/+}* males (expressing *Cre* in the germline, http://www.jax.org/strain/024098) with wild-type FVB females. *mGFAP-Cre;Smarcb1^{del/flox}* mice were obtained by breeding *mGFAP-Cre; Smarcb1^{flox/+}* males with *Smarcb1^{flox/flox}* females. Mice were genotyped by PCR amplification of DNA extracted from tail biopsies using the primers cre-s1 (5′-ACA TGT TCA GGG ATC GCC AG-3′) and cre-a1 (5′-TAA CCA GTG AAA CAG CAT TGC-3′) for the *Cre* transgene, 407Fbis (5′-GGA AAA TCT AGA AAG CAC AAA TGA-3′) and 408R (5′-TGT AGT CTA GGC TGG GTG TG-3′) for the *Smarcb1^{WT}* and *Smarcb1^{flox}* alleles, 407Fbis and 416rbis (5′-CCT GGG GCA GCT CTC TAC A-3′) for the *Smarcb1^{del}* allele, NF2Flox2-S (5′-CTT CCC AGA CAA

GCA GGG TTC-3′) and NF2Flox2-A (5′-GAA GGC AGC TTC CTT AAG TC-3′) for the $Nf2^{WT}$ and $Nf2^{flox}$ alleles and Z1 (5′-GCG TTA CCC AAC TTA ATC G-3′) and Z2 (5′-TGT GAG CGA GTA ACA ACC-3′) for the *AZCL* transgene.

**Magnetic resonance imaging.** Mice were anesthetized with isoflurane throughout the imaging procedure. Mice were inserted in the prone position into a small animal MRI scanner (PharmaScan 300; Bruker BioSpin Division, Billerica, MA, USA) 7 Tesla magnet using the 19-mm inner diameter transmit receive coil. ParaVision 4.0 scanner software (BRUKER BioSpin MRI GmbH, Ettlingen, Germany) was set to use Rapid Acquisition with Relaxation Enhancement (RARE) spin echo sequence for fast T2-weighted imaging (TE 50, TR 3000, RARE Factor 8) with a $256 \times 256$ in-plane matrix and a 2.56-cm field of view. After scanning, if needed, mice were gently warmed on a thermostatically controlled heating pad until awake enough to be returned to their home cage. MRI images were reconstructed at native resolution and analyzed with OsiriX (Open-Source Software for Navigating in Multidimensional DICOM Images) image analysis software packages.

**H&E staining, immunohistochemistry, X-gal staining and cell counting.** Hematoxylin and eosin staining was performed on 3.5-μm-thick sections prepared from paraffin blocks of formalin-fixed tissues. Immunohistochemistry was performed on 9 *P0-CreC;Smarcb1flox/flox* representative tumors (3 tumors per histological pattern) and 7 representative schwannomas from *P0-CreC;Nf2flox/flox* mice (Fig. 3 and Supplementary Table 1). A standard protocol was used with primary antibodies incubated 1 h at room temperature: cytokeratin (1/500, Dako, M3515), FABP7 (1/1000, Abcam, ab32423), GFAP (1/3000, Dako, Z0334), Ki-67 (1/1000, BD Pharmingen, 556003), merlin (1/400, Cell Signaling Technology, #6995), neurofilament triplet proteins (1/500, Enzo Life Sciences, BML-NA1223), PGDS (1/1000, Santa Cruz Biotechnology, sc-14826), S100 (1/10000, Dako, Z0311), SMA (1/200, BioCare Medical, CME 305), SMARCB1 (1/200, BD Transduction Laboratories, 612110), vimentin (1/200, Cell Signaling Technology, #5741). Biotinylated secondary anti-rabbit, anti-mouse (Vector Laboratories, Burlingame, CA) or anti-goat (Dako) IgG antibodies were incubated 30 min at room temperature. Immunoreactivity was semi-quantitatively scored based on the percentage of positive tumor cells (few, foci or majority of positive cells) and the intensity of staining (weak, moderate or strong).

To assess cell proliferation, bromodeoxyuridine (50 mg/kg body weight) was administrated 4 h before dissection of the tissues. Incorporation of BrdU was then revealed by immunohistochemistry with BrdU antibody (Fitzgerald Industries International, 20-BS17) and secondary anti-sheep antibody (Jackson ImmunoResearch Laboratories, West Grove, PA).

For X-gal staining, tissues were frozen in OCT compound (Tissue-Tek, 4583) on dry ice. Frozen sections of 10 μm were fixed in 2% formaldehyde and 0.2% glutaraldehyde in 1X PBS for 10 min at room temperature. Fixed tissues were washed three times in 1X PBS, and incubated overnight at 37 °C in the staining solution (5 mM $K_3[Fe(CN)_6]$, 5 mM $K_4[Fe(CN)_6]$, 2 mM $MgCl_2$ and 2 mg/ml X-gal in PBS). After staining, tissues were washed twice in 1× PBS and counterstained with Nuclear Fast Red for 1–2 min.

For DRG cell count, 5–12 DRGs of the cervical and/or thoracic spine were selected from 3–5 mice per genotype using a light microscope Axio Imager M1 (Zeiss). The region of interest (ROI) containing ganglion cells was manually demarcated on captured images of 20× magnification fields. The Axiovision software was used to determine the surface and the cells count in the ROI. All histopathological analysis was performed in blind under the guidance of an experienced pathologist.

**PCR.** PCR amplification was performed on DNA extracted from multiple tissues using the primers cre-s1 and cre-a1 for the *Cre* transgene, 407Fbis and 408R for the *Smarcb1WT* and *Smarcb1flox* alleles, 407Fbis, 408R, and 416R (5′-GCC ACC AGC CAG ATG TCA TAC-3′) for the *Smarcb1flox* and *Smarcb1del* alleles in the tumors, Ini1DelAf (5′-AAG CAC AAA TGA GAG AAA ACG TA-3′) and Ini1DelBr (5′-TGC CAC CAG CCA GAT GTC A-3′) with Hotmaster Taq DNA polymerase for the *Smarcb1del* allele in the PCR for all the tissues.

**RNA-Seq and bioinformatics.** RNA was isolated using the RiboPure kit (Life Technologies) according to manufacturer's instructions. RNA Libraries were prepared using the TruSeq Stranded library preparation kit (Illumina) RNA after RiboZero treatment for removal of ribosomal RNAs. After library preparation, amplified double-stranded cDNA was fragmented into 125 bp (Covaris-S2, Woburn, MA) DNA fragments, which were (200 ng) end-repaired to generate blunt ends with 5′- phosphates and 3′- hydroxyls and adapters ligated. The purified cDNA library products were evaluated using the Agilent Bioanalyzer (Santa Rosa, CA) and diluted to 10 nM for cluster generation in situ on the HiSeq paired-end flow cell using the CBot automated cluster generation system. All samples were multiplexed into a single pool in order to avoid batch effects and sequenced using an Illumina HiSeq 4000 sequencer (Illumina, San Diego, CA) yielding between 52 and 90 million reads per sample. Quality control was performed on base qualities and nucleotide composition of sequences. Alignment to the M. musculus (mm10) refSeq (refFlat) reference gene annotation was performed using the STAR spliced

read aligner with default parameters. Additional QC was performed after the alignment to examine: the level of mismatch rate, mapping rate to the whole genome, repeats, chromosomes, key transcriptomic regions (exons, introns, UTRs, genes), insert sizes, AT/GC dropout, transcript coverage and GC bias. Between 80 and 85% (average 83%) of the reads mapped uniquely to the mouse genome and only uniquely mapped reads were used for subsequent analyses. Total counts of read-fragments aligned to candidate gene regions were derived using HTSeq program (version 0.6.0, www.huber.embl.de/users/anders/HTSeq/doc/overview.html) with mouse mm10 refSeq (refFlat table) as a reference and used as a basis for the quantification of gene expression.

Microarray data from Johann et al.[43] (GSE70678), Han et al.[30] (GSE64019) and Ng et al.[44] (GSE68627) was downloaded from GEO and normalized using the robust multiarray method (RMA). Gene symbols were used to match human and mouse transcripts and merge expression datasets after averaging multiple probes querying the same gene. Batch effects were corrected using ComBat and overall quantile normalization was performed across data sets.

Hierarchical clustering of the top most variable genes (as ranked by s.d.) was performed using the plotDendroAndColors function from the WGCNA package. Weighted Gene Co-expression Network Analysis (WGCNA) was conducted using the R package. Briefly, correlation coefficients were constructed between expression levels of genes, and a connectivity measure (topological overlap, TO) was calculated for each gene by summing the connection strength with other genes. Genes were then clustered based on their TO, and groups of co-expressed genes (modules) were identified. Each module was assigned a color, and the first principal component (eigengene) of a module was extracted from the module and considered to be representative of the gene expression profiles in a module. The phenotypic trait of interest is then regressed on the eigengene to examine whether there was a significant relationship between the module and the trait.

The optimal number of clusters was determined using the non-negative matrix factorization (NMF) method with the NMF R package, based on the top 5000 most variable genes with standard NMF algorithm method (brunet) performed on 10 runs. The optimal numbers of clusters have been defined based on the maximum cophenetic correlation score.

**Quantitative RT-PCR.** Reverse transcription was performed with 1 μg of the same RNA samples used for RNA-Seq and the SuperScript IV VILO Master Mix (Life Technologies) following the manufacturer's instructions. Gene expression levels were assessed by quantitative PCR using a QuantStudio 3 real-time PCR system (Applied Biosystems) and TaqMan gene expression assays: Smarcb1 (Mm00448776_m1), Ezh2 (Mm00468464_m1) and Hprt (Mm00446968_m1, Life Technologies). Relative expression was determined using the $2^{-\Delta\Delta Ct}$ method with Hprt expression as reference and normalized to a brain control sample.

**Western blot.** Tissues were snap-frozen in liquid nitrogen and stored at −80 °C. Pulverized frozen tumor samples were lysed in radioimmunoprecipitation assay (RIPA) buffer and the total proteins were extracted, separated, and transferred using standard procedures. Antibodies were purchased from BD Transduction Laboratories: SMARCB1 (1/500, 612110); Cell Signaling Technology: Akt (1/2000, #9272), P-Akt (Ser473) (1/2000, #4060), CDK4 (1/2000, #2906), CDK6 (1/2000, #3136), Cyclin D1 (1/3000, #2926), merlin (1/1000, #6995), p27 (1/1000, #2552), P-Rb (1/1000, #8516), S6 (1/1000, #2317), P-S6 (Ser 235/236) (1/4000, #4858), 4E-BP1 (1/2000, #9452); Novus Biologicals: GAPDH (1/5000, NB300-221), PTCH1 (1/1000, MAB41051), RB1 (1/50, NB120-3077), SHH (1/500, AF464); R&D Systems: GLI-1 (1/1000, AF3455); Santa Cruz Biotechnology: p16 (1/3000, sc-1207) and Thermo Scientific: p21WAF1 (1/1000, MS-387-P0). All uncropped scans of western blots are provided in Supplementary Fig. 7.

**Statistical analysis.** GraphPad Prism software version 5.0 (GraphPad Software, San Diego, CA) was used to perform statistical analysis and construct Kaplan–Meier survival curves. Mendelian birth incidence of mouse genotypes was examined using chi-square test. Differences in survival of mouse genotypes were analyzed by log-rank Mantel–Cox test. Two-tailed Student's unpaired $t$-test was used to determine significance between groups unless otherwise indicated. The variance between the groups was not significantly different (F test). A P value of < 0.05 was considered statistically significant. Results of the significance tests are reported as follow: ns (non significant), *($P < 0.05$), **($P < 0.01$) and ***($P < 0.001$). Error bars represent s.e.m.

**Data availability.** The RNA-Sequencing data presented in this study have been deposited in NCBI Gene Expression Omnibus with the accession number GSE94082. All other data supporting the findings of this study are available in Supplementary Information Files and from the corresponding author upon reasonable request.

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

## Acknowledgements

We would like to thank Dr. Anat Stemmer-Rachamimov (Massachusetts General Hospital and Harvard Medical School, Boston, MA) for histopathological review of mouse schwannomas; Rosa Sierra, Fabrice Chareyre, Rocky Adams and Benedicte Chareyre for technical support; Pr. Michel Kalamarides (Groupe Hospitalier Pitié-Salpêtrière, Paris, France) for meningeal RT immunostainings; Gevorg Karapetyan (Children's Hospital, Los Angeles, CA) for mouse MRI services; Dr. Agnes Klonchendler (The Hebrew University of Jerusalem, Jerusalem, Israel) for providing the *Smarcb1^flox* mouse strain; Dr. Dies N Meijer (Erasmus University Medical Center, Rotterdam, Netherlands) for providing the *DHH-Cre* mouse strain; Dr. Michael V. Sofroniew (University of California Los Angeles, Los Angeles, CA) for providing the *mGFAP-Cre* mouse strain; Dr. Jaclyn A. Biegel for invaluable input on the project. This work was supported by the Children's Tumor Foundation Schwannomatosis awards No. 2008-02-007 and 2011-02-010 (M.G.); U.S. Army Medical Research and Materiel Command, through the Neurofibromatosis Research Program under Award No. W81XWH-10-1-0070 (J.V.); the NINDS Informatics Center for Neurogenetics and Neurogenomics P30 NS062691 (G.C.); the House Research Institute and the Department of Head and Neck Surgery at UCLA. Opinions, interpretations, conclusions and recommendations are those of the author and are not necessarily endorsed by the Department of Army.

## Author contributions

J.V. designed the study, performed experiments, analyzed data and wrote the manuscript. F.G. and G.C. performed the bioinformatics analyses. A.R.J. interpreted the histological data and revised the manuscript. M.G. designed and supervised the study and wrote the manuscript.

## Additional information

**Competing interests:** The authors declare no competing financial interests.

