## [Peer Review File · Nature Communications]

Reviewers' comments:

Reviewer #1 (Remarks to the Author): Expert in AT/RT

The manuscript by Vitte and colleagues addresses the issue of modeling schwannomas and rhabdoid tumors in mice, two tumors with very different aggressiveness that are both associated with mutations of SMARCB1.

They show that P0-driven inactivation of *Smarcb1* leads to the high penetrance occurrence of rhabdoid-like tumors in the cranial nerves and possibly in the meninges. In contrast, tumors are not observed when the inactivation of *Smarcb1* is induced later in neural crest cell development. They have also generated double GFAP-driven *Smarcb1* and *Nf2* inactivation. These mice develop Schwannomas that are similar to the ones observed in single *Nf2* mutant mice.

This work constitutes undoubtedly an interesting new model to investigate the oncogenic role of *Smarcb1* inactivation but, rather unfortunately, these double mutant mice develop tumors that are similar to the single *NF2* mutant mice and therefore fail to model the 4-hits/3-steps hypothesis. While this manuscript relies on an impressive amount of work, some major concerns need to be addressed.

1- Though the *Smarcb1*^{-/-} tumors they observed have phenotypic similarities with RTs it must also be noted that the tumor localization (mostly cranial nerves) are relatively unusual for human RTs even though sporadic case reports of such localizations (thoroughly quoted by the authors) have been published. This modeling of a relatively narrow spectrum of the human malignancy should be more precisely indicated and discussed.

2- The extensive IHC analyses are interesting but they still only investigate a small subset of markers. A more comprehensive expression profiling should be undertaken. This may allow performing a more thorough comparison with the numerous expression profiles that have been published for human and mouse *Smarcb1*-deficient tumors and that are available in different databases. In particular recent data shows that different subgroups of RTs exist in humans and in mice. Such a profiling may enable a more precise identification of the corresponding human and mouse subgroup(s).

3- The title and abstract are somehow misleading. In fact the authors fail to develop a model of *Smarcb1*-induced Schwannomas as the double *Smarcb1*/*NF2* mutants do not behave differently than the single *NF2* mutant hence providing no clue to elucidate the respective roles of these two genes in schwannoma development.

4- The authors should be more cautious concerning the impact of their work on the identification of the RT cell-of-origin. They indeed show that neural crest cells can be transformed by *Smarcb1* inactivation hence suggesting that a minority of RTs (see above comment) may have such a cell-of-origin but this should not lead to a broad conclusion that this neural crest cell progenitor is, in general, the RT cell-of-origin.

5- The references need to be reviewed. Some important landmark papers on RTs are not quoted while the authors do reference a high number of relatively marginal reports.

6- The suggestion that the observed tumors correspond to the recently described SHH group is only supported by the expression of *Gli1* which constitutes a relatively weak proof. Again, comprehensive transcriptome data and/or a clear demonstration of a nuclear localization of *Gli1* would be necessary to further document this hypothesis.

7- Fig.8 is misleading as single *Nf2*^{-/-} mice which behave as double *Smarcb1*/*Nf2* mutant mice are not indicated. The authors cannot draw conclusions on the 4-hits/3-steps model.

8- Fig7 shows that whorls of *Baf47* negative cells are only observed in the *mGFAP*-cre; *Nf2*^{-/-}; *Smarcb1*^{-/-} and not in the *mGFAP*-cre; *Smarcb1*^{-/-} background. May this indicate that in this context the loss of *Nf2* is necessary for the survival of *Smarcb1* cells?

Reviewer #2 (Remarks to the Author): Expert in AT/RT

The manuscript by Vitte et al. seeks to elucidate the timing of genetic events leading to

schwannoma and a rare cancer called rhabdoid tumor.

The presented data is novel and explains long observed findings and the link between two completely different diseases.

The manuscript is well structured and written and especially the pictorial part helps to understand the experimentation and gives in the end an excellent overview of potential events. The application of the 4-hit/3-step model nicely fits the authors line of evidence.

There is only very limited criticism:

1) the functional consequences of inactivation of the genes SMARCB1 and NF2 is mainly described in morphological terms. The results of some expression data is presented in Figure 5. However the consequences of gene inactivation of SMARCB1 and NF2 on the SWI/SNF complex and its remaining members is not addressed at all. It would be very interesting to see whether the complex loses all of its functions and composition or whether other mechanisms are responsible for the morphological change. It might be worthwhile to try and elucidate gene set enrichment in the different constructs. The limited data and discussion on molecular subtyping (shh) is a step in the right direction.

2) the authors should be encouraged to discuss a few findings in human patients in relation to their data. How for example would they explain the phenomenon of synchronous tumors? While the authors found Cre and SMARCB1 deletion in kidneys one of the major target organs of rhabdoid tumors, they did not see any tumors in the kidney. Furthermore there are some human families in which family members are affected by rhabdoid tumors while others have schwannomas. How can this data be reconciled?

3) As the presented mouse models are only the second model of rhabdoid tumors which consistently demonstrate intracranial RT without any other sign of malignancy outside the CNS the manuscript would benefit from a closer comparison to the recent manuscript by Han et al (Nat Comm 2016).

4) On a minor note it would be nice to see a proliferation marker when examining the mouse tumors (e.g. KI67 or mouse equivalent).

5) very minor problems are on page 20 line 411 it should read "a cell of origin".

The references are at times incomplete and not consistently formatted e.g. Ref 66 misses page numbers and edition numbers. Ref 47 should be exchanged for the new WHO classification published a few months ago. All references should be checked for correctness.

All in all this is a beautiful manuscript with novel and important findings with interest not only for the specialist, but also for any researcher interested in the mechanisms of disease. Especially the influence of spatial and temporal factors is very interesting and has a potential to help understand many disease states.

I thus suggest to accept the manuscript following minor revisions.

Reviewer #3 (Remarks to the Author): Expert in Schwannomas

In this manuscript, Vitte et al examine the question of why SMARCB1 mutations result in the development of malignant rhabdoid tumors in some patients and benign schwannomas in others; answering this question required that they look both at whether there were windows of susceptibility that differed for these two tumor types and address the question of the cellular origin of rhabdoid tumors (which has been controversial up to now). To test the hypothesis that Smarcb1 inactivation in neural crest cells and the Schwann cell lineage is sufficient for schwannoma

pathogenesis, the authors generated PO-CreC;Smarcb1flox/flox mice. They report that 65% of these mice developed aggressive tumors that resembled human rhabdoid tumors. In contrast, PO-CreB;Smarcb1flox/flox mice mostly died in utero or perinatally; those that did survive, died between 1.2-2 months of age with retroorbital or nasal cavity tumors. In contrast, Dhh-Cre;Smarcb1flox/flox mice did not develop tumors. GFAP-Cre;Smarcb1flox/flox mice developed tumors with rhabdoid features between 2 and 17 months of age. In contrast, GFAP-Cre;Smarcb1flox/flox;Nf2flox/flox mice developed schwannomas. Based on these observations, the authors argue that Smarcb1 loss in neural crest cells early in development leads to rhabdoid tumor formation, while Smarcb1 loss at a later developmental stage combined with Nf2 loss results in schwannoma pathogenesis.

The topic addressed in this manuscript is an important one and potentially of strong interest to the readership of Nature Communications. However, in its current form, the manuscript is very difficult to follow and the arguments presented in support of their conclusions are convoluted and not clearly supported by the data as currently presented. Points that the authors need to address to improve the manuscript are as follows:

- 1) Although the data in the manuscript are quite interesting, there are a number of sentences in the manuscript that are awkwardly phrased that make it difficult to follow the manuscript. For example, there is a sentence in the abstract that states "The single loss of Smarcb1 in neural crest cells is sufficient to initiate tumorigenesis...". The Results section in particular is very dense and difficult to follow; while this is in part, this is due to the complicated genotypes being presented, it also reflects the often decipherable sentences in this section of the paper. The manuscript would be much easier to follow if the authors thoroughly edited it so as to present their work in a more straightforward and less complicated manner.
- 2) A major theme of this manuscript is that the timing of Smarcb1 loss (and likely the stage at which this loss occurs during neural crest/Schwann cell development) dictates whether the mouse develops a schwannoma or a rhabdoid tumor. This argument is dependent upon knowing the timing with which the different Cre transgenes they use are expressed in their mouse models. However, with the exception of a statement very late in the Results (when discussing the GFAP-Cre driver), the authors provide no information in the Results about the times when these different Cre transgenes are expressed. This is finally presented in Figure 8, which is first referred to in the next to last paragraph of the Discussion. As a Result, I found myself struggling to make sense of their data (and argument) until the very end of the paper. The authors need to clearly indicate when each of these transgenes are active in the embryo when they first present the Cre driver to the reader.
- 3) In the first paragraph of the Results, the authors mention that some of the PO-CreC;Smarcb1flox/flox mice had bloated intestines with an emaciated wall and a reduced number of villi. Why did this occur?
- 4) The authors had their tumors examined by competent pathologists whose opinions I respect. However, many of the journal's readers will likely not know these individuals. Given that, the authors really need to do a better job of presenting the diagnostic criteria that were applied and a critical assessment of how their mouse tumors compared to their human counterparts. There clearly were some differences—for instance, the authors simply stated that their mouse rhabdoid tumors were negative for neurofilament and cytokeratin. Human AT/RTs commonly show foci with expression of both of these antigens and the authors made no mention of this.
- 5) The authors note that the PO-CreB transgene expresses Cre "in a larger cell compartment" than the PO-CreC transgene. Again, what does this mean? Is the timing of expression also different from PO-CreC? This is not mentioned in the text and it is not indicated in Figure 8.
- 6) It is not clear what the authors mean by the sentence that spans lines 200-203. If part of this sentence is trying to argue that the tumors could have originated from the leptomeninges, despite the absence of PGDS, it is unclear how the authors arrived at this possibility.
- 7) Why is the data included on the DHH-Cre;Smarcb1flox/flox mice (they didn't develop tumors)? Are they trying to argue that these mice help them narrow down the window when Smarcb1 loss results in rhabdoid tumor formation? That possibility seems unlikely, given that GFAP-

Cre;Smrbc1flox/flox mice developed tumors with rhabdoid features between 2 and 17 months of age (this Cre driver turns on at E13.5 compared to Dhh-Cre, which turns on at E12). The significance of these negative data are unclear as currently presented.

8) Panel e in Figure 2 is not convincing and needs to be presented with a better image. In addition, the tumor images presented in Figure 2 are at very low power. Following the usual pathology dictum that "blue is bad", it seems likely that these are aggressive tumors, but it will not be possible to be sure that this is the case until higher power images are included.

Point-to-point response to Reviewers' comments:

All changes in the revised manuscript are highlighted in yellow in the marked-up text file.

Reviewer #1 (Remarks to the Author): Expert in AT/RT

... While this manuscript relies on an impressive amount of work, some major concerns need to be addressed.

*1- Though the *Smarcb1*^{-/-} tumors they observed have phenotypic similarities with RTs it must also be noted that the tumor localization (mostly cranial nerves) are relatively unusual for human RTs even though sporadic case reports of such localizations (thoroughly quoted by the authors) have been published. This modeling of a relatively narrow spectrum of the human malignancy should be more precisely indicated and discussed.*

Introduction (line 72) was changed to: "and sporadic case reports of RTs emanating from cranial nerves"

The conclusion paragraph in the discussion (lines 466-472) was changed to: "In conclusion, we have shown that biallelic inactivation of *Smarcb1* in the neural crest leads to the development of PNS and meningeal RTs, thus identifying the cell of origin for these RT subsets in an early neural crest population."

*2- The extensive IHC analyses are interesting but they still only investigate a small subset of markers. A more comprehensive expression profiling should be undertaken. This may allow performing a more thorough comparison with the numerous expression profiles that have been published for human and mouse *Smarcb1*-deficient tumors and that are available in different databases. In particular recent data shows that different subgroups of RTs exist in humans and in mice. Such a profiling may enable a more precise identification of the corresponding human and mouse subgroup(s).*

We performed RNA-sequencing to compare gene expression profiles of RTs found in our *P0-CreC;Smarcb1^{flox/flox}* mice and previously published human RTs (Johann *et al.* 2016), *Rosa26-Cre^{ERT2};Smarcb1^{flox/flox}* (Han *et al.* 2016) and *hGFAP-Cre;Smarcb1^{flox/flox};Trp53^{flox/flox}* (Ng *et al.* 2015) *Smarcb1*-deficient models. We found that tumors from *P0-CreC;Smarcb1^{flox/flox}* mice clustered in the three molecular subgroups recently defined by Johann *et al.*, thus showing that our mouse model accurately reproduces the molecular diversity found in human RTs.

Moreover, the comparison with two existing mouse models (Han *et al.*, Ng *et al.*) showed that mouse RTs clustered in three molecular subgroups irrespective of the model of origin. All these data were added to the Results and Discussion sections.

*3- The title and abstract are somehow misleading. In fact the authors fail to develop a model of *Smarcb1*-induced Schwannomas as the double *Smarcb1/NF2* mutants do not behave differently than the single *NF2* mutant hence providing no clue to elucidate the respective roles of these two genes in schwannoma development.*

We think that the crucial point here is that in *mGFAP-Cre* mice the double *Smarcb1/Nf2* mutants

(schwannoma +) behave differently than the single *Smarcb1* mutant (schwannoma -).

The observation of frequent somatic, tumor-specific *NF2* mutations, and the loss of the second *NF2* allele in schwannomas from patients with germline *SMARCB1* mutations (Boyd et al. 2008; Sestini et al. 2008; Hadfield et al. 2008, 2010) strongly suggests that the classical two-hit model of tumorigenesis does not pertain in the tumors of schwannomatosis patients, at least in the sense that it would require biallelic *SMARCB1* inactivation to be sufficient for tumor initiation or growth (Plotkin et al. 2013, Kehrer-Sawatzki et al. 2016). Our hypothesis is that in schwannomatosis patients the role of *SMARCB1* loss in schwannomagenesis is to promote and facilitate biallelic *NF2* loss (through the 4-hit/3-step mechanism), and that *NF2* loss is necessary and sufficient for schwannoma formation. Since the genetic 4-hit/3-step mechanism of double *NF2*/*SMARCB1* inactivation cannot be reproduced in mice because of the two genes being located in different chromosomes, we used conditional mutagenesis to model the different steps.

In line with this hypothesis, the fact that *mGFAP-Cre;Smarcb1^{fllox/fllox}* mutant mice do not develop schwannoma and that *mGFAP-Cre;Smarcb1^{fllox/fllox};Nf2^{fllox/fllox}* double mutant mice develop benign schwannomas reinforces and models, for the first time, the necessary role of *Nf2* loss for schwannoma formation in mice and schwannomatosis patients.

Finally, the fact that *mGFAP-Cre;Smarcb1^{fllox/fllox};Nf2^{fllox/fllox}* double mutant mice and *mGFAP-Cre;Nf2^{fllox/fllox}* single mutant mice develop schwannomas is again consistent with the human condition where both *NF2* and schwannomatosis patients develop benign, grade I schwannomas.

Details on the temporal window of susceptibility to RT have been added to the Results (lines 331-333) and Discussion (lines 403-404) sections. The Abstract was modified to reflect the necessity of the early *Smarcb1* loss in inducing RTs.

The reason for including “*Smarcb1* and *Nf2* inactivation” in the title is to stress the fact that individually both genes are crucial in determining the development of RT or schwannoma.

4- The authors should be more cautious concerning the impact of their work on the identification of the RT cell-of-origin. They indeed show that neural crest cells can be transformed by Smarcb1 inactivation hence suggesting that a minority of RTs (see above comment) may have such a cell-of-origin but this should not lead to a broad conclusion that this neural crest cell progenitor is, in general, the RT cell-of-origin.

Introduction and Discussion sections were modified according to this comment.

See also point #1.

5- The references need to be reviewed. Some important landmark papers on RTs are not quoted while the authors do reference a high number of relatively marginal reports.

As suggested by the reviewer, we have revised the choice of references to include landmark references on RTs.

6- The suggestion that the observed tumors correspond to the recently described SHH group is only supported by the expression of Gli1 which constitutes a relatively weak proof. Again, comprehensive

transcriptome data and/or a clear demonstration of a nuclear localization of Gli1 would be necessary to further document this hypothesis.

We have included a comprehensive molecular analysis of RTs obtained in the *P0-CreC;Smarb1^{fllox/fllox}* mouse model by whole genome RNA-sequencing (12 representative tumors).

See answer to point #2.

7- Fig.8 is misleading as single Nf2^{-/-} mice which behave as double Smarb1/Nf2 mutant mice are not indicated. The authors cannot draw conclusions on the 4-hits/3-steps model.

mGFAP-Cre;Nf2^{fllox/fllox} mice were added to now Fig. 7.

Indeed, the original text did not stress adequately that our model does not reproduce the 3 genetic steps of double *SMARCB1/NF2* inactivation sequentially in a single mouse (*Smarb1* and *Nf2* genes are not syntenic in the mouse). Instead, we generated mice for each of the 4 possible gene hit combinations to dissect the impact of each combination on tumorigenesis.

See also point #3.

8- Fig7 shows that whorls of Baf47 negative cells are only observed in the mGFAP-cre; Nf2^{-/-}; Smarb1^{-/-} and not in the mGFAP-cre; Smarb1^{-/-} background. May this indicate that in this context the loss of Nf2 is necessary for the survival of Smarb1 cells?

Comparison of SMARCB1 staining in DRG of control and *mGFAP-Cre;Smarb1^{fllox/fllox}* mice (now Fig. 6) shows unstained nuclei in the *mGFAP-Cre;Smarb1^{fllox/fllox}* mice. As SMARCB1-negative cells in *mGFAP-Cre;Smarb1^{fllox/fllox}* DRG don't proliferate, they are present but in smaller number than in *mGFAP-Cre;Smarb1^{fllox/fllox};Nf2^{fllox/fllox}* DRG, thus explaining why they are not easily visible. For more clarity, we added insets with higher magnification of representative SMARCB1-negative cells in *mGFAP-Cre;Smarb1^{fllox/fllox}* DRG and SMARCB1-positive cells in control DRG.

Reviewer #2 (Remarks to the Author): Expert in AT/RT

... The presented data is novel and explains long observed findings and the link between two completely different diseases.

The manuscript is well structured and written and especially the pictorial part helps to understand the experimentation and gives in the end an excellent overview of potential events. The application of the 4-hit/3-step model nicely fits the authors line of evidence.

There is only very limited criticism:

1) the functional consequences of inactivation of the genes SMARCB1 and NF2 is mainly described in morphological terms. The results of some expression data is presented in Figure 5. However the consequences of gene inactivation of SMARCB1 and NF2 on the SWI/SNF complex and its remaining members is not addressed at all. It would be very interesting to see whether the complex

loses all of its functions and composition or whether other mechanisms are responsible for the morphological change. It might be worthwhile to try and elucidate gene set enrichment in the different constructs. The limited data and discussion on molecular subtyping (shh) is a step in the right direction.

We performed RNA-sequencing to assess the gene expression profile of the *P0-CreC;Smarchb1^{fllox/fllox}* tumors and compare these tumors with published human AT/RT molecular subgroups (Johann *et al.* 2016) and other published *Smarchb1*-deficient mouse models (Han *et al.* 2016; Ng *et al.* 2015).

We found that RTs from *P0-CreC;Smarchb1^{fllox/fllox}* mice clustered among the three molecular subgroups recently defined by Johann *et al.*, thus showing that our mouse model accurately reproduces the molecular diversity found in human RTs. The comparison with two existing mouse models (Han *et al.*, Ng *et al.*) showed that mouse RTs clustered in three molecular subgroups irrespective of the model of origin. All these data were added to the Results and Discussion sections.

Analysis of SWI/SNF complex components demonstrated that tumors from *P0-CreC;Smarchb1^{fllox/fllox}* mice specifically expressed subunits found in the pluripotent embryonic stem cell (esBAF) and in the multipotent neural progenitors (npBAF) complexes. Except for subunit CREST (*SS18L1* gene), this results was consistent in human AT/RT samples from Johann *et al.*

2) the authors should be encouraged to discuss a few findings in human patients in relation to their data. How for example would they explain the phenomenon of synchronous tumors?

Synchronous tumors usually develop in patients with cancer predisposition syndromes and are consistent with a germline (or de novo mutation) and a second genetic hit (chromosome 22 complete or partial loss resulting in the loss of the normal *SMARCB1* gene) occurring in different tissues at an early stage of development.

This phenomenon has been described in few patients with RTs and carrying *SMARCB1* mutations (Biegel *et al.* 2000, Seeringer *et al.* 2014, Metts *et al.* 2017). As stated in the Results, 62.5% of *P0-CreC;Smarchb1^{fllox/fllox}* mice displayed multiple tumors at the time of dissection. We added this comment to the Discussion section (lines 448-450).

While the authors found Cre and SMARCB1 deletion in kidneys one of the major target organs of rhabdoid tumors, they did not see any tumors in the kidney.

In addition to results in this manuscript (Supplementary Fig. 1) we have previously shown (Giovannini *et al.* 2000) Cre expression in the kidney of *P0-CreB* and *P0-CreC* mice and Asada *et al.* (2011) using another *P0-Cre* mouse line showed that most of the kidney fibroblasts derived from neural crest precursors. Thus, the fact that no tumors are found in *P0-CreC;Smarchb1^{fllox/fllox}* mice can be explained by two hypotheses: Cre in *P0-CreC* and *P0-CreB* mice is not expressed in the correct cell of origin of kidney RTs or is not expressed during the right temporal window of sensitivity to tumorigenesis.

This point was clarified in the Results section (lines 149-151).

Furthermore there are some human families in which family members are affected by rhabdoid

tumors while others have schwannomas. How can this data be reconciled?

The existence of adult carriers of a *SMARCB1* mutation without RT in families presenting either RTs or schwannomatosis suggests that there is a specific developmental time window during which RT progenitor cells are vulnerable to *SMARCB1* loss (Swensen *et al.* 2009, Eaton *et al.* 2011, Carter *et al.* 2012).

In this model, if loss of the second *SMARCB1* allele occurs during early development, the individual will develop a RT. The *P0-CreC;Smarchb1^{flox/flox}* mice mimic this situation.

If mutation of the second *SMARCB1* allele does not occur during early development, the individual carrying a germline *SMARCB1* mutation does not develop a RT. This situation is modeled in our *mGFAP-Cre;Smarchb1^{del/flox}* mice that do not develop RT (late Cre expression inducing late *Smarchb1* loss).

The *SMARCB1* mutation carrier can then develop schwannomatosis at a later age when the additional somatic mutation steps occur: loss of *SMARCB1* 2nd allele in addition to loss of 1st *NF2* allele (2nd step) and the second *NF2* mutation (3rd step). This combination of *SMARCB1* and *NF2* loss is modeled in *mGFAP-Cre;Smarchb1^{flox/flox};Nf2^{flox/flox}* mice that develop schwannomas.

This point was added to the Results (lines 292-294) and Discussion sections (lines 404-406).

3) As the presented mouse models are only the second model of rhabdoid tumors which consistently demonstrate intracranial RT without any other sign of malignancy outside the CNS the manuscript would benefit from a closer comparison to the recent manuscript by Han et al (Nat Comm 2016).

We performed gene expression analysis by RNA-Sequencing of 12 rhabdoid tumors from the *P0-CreC;Smarchb1^{flox/flox}* mouse model. Unsupervised clustering analysis demonstrated that the tumors from Han et al. mouse model clustered with two of the three molecular subgroup found in our *P0-CreC;Smarchb1^{flox/flox}* mouse model. This is consistent with Han et al. analysis where they show that their tumors clustered in two subgroups. These data are now shown in Fig. 4b.

4) On a minor note it would be nice to see a proliferation marker when examining the mouse tumors (e.g. KI67 or mouse equivalent).

The Ki-67 proliferation marker was used and demonstrated a high proliferative index in the RTs from *P0-CreC;Smarchb1* mice (Fig. 3f) and very low proliferation in the schwannomas from *mGFAP-Cre;Smarchb1^{flox/flox};Nf2^{flox/flox}* and *mGFAP-Cre;Nf2^{flox/flox}* mice (Fig. 6, last row).

5) very minor problems are on page 20 line 411 it should read "a cell of origin".

The references are at times incomplete and not consistently formatted e.g. Ref 66 misses page numbers and edition numbers. Ref 47 should be exchanged for the new WHO classification published a few months ago. All references should be checked for correctness.

These items were corrected.

All in all this is a beautiful manuscript with novel and important findings with interest not only for the specialist, but also for any researcher interested in the mechanisms of disease. Especially the influence of spatial and temporal factors is very interesting and has a potential to help understand many disease states.

I thus suggest to accept the manuscript following minor revisions.

Reviewer #3 (Remarks to the Author): Expert in Schwannomas

...The topic addressed in this manuscript is an important one and potentially of strong interest to the readership of Nature Communications. However, in its current form, the manuscript is very difficult to follow and the arguments presented in support of their conclusions are convoluted and not clearly supported by the data as currently presented. Points that the authors need to address to improve the manuscript are as follows:

1) Although the data in the manuscript are quite interesting, there are a number of sentences in the manuscript that are awkwardly phrased that make it difficult to follow the manuscript. For example, there is a sentence in the abstract that states “The single loss of Smarcb1 in neural crest cells is sufficient to initiate tumorigenesis...”. The Results section in particular is very dense and difficult to follow; while this is in part, this is due to the complicated genotypes being presented, it also reflects the often decipherable sentences in this section of the paper. The manuscript would be much easier to follow if the authors thoroughly edited it so as to present their work in a more straightforward and less complicated manner.

These points were extensively revised with a more straightforward comparison of mouse vs. human phenotypes.

2) A major theme of this manuscript is that the timing of Smarcb1 loss (and likely the stage at which this loss occurs during neural crest/Schwann cell development) dictates whether the mouse develops a schwannoma or a rhabdoid tumor. This argument is dependent upon knowing the timing with which the different Cre transgenes they use are expressed in their mouse models. However, with the exception of a statement very late in the Results (when discussing the GFAP-Cre driver), the authors provide no information in the Results about the times when these different Cre transgenes are expressed. This is finally presented in Figure 8, which is first referred to in the next to last paragraph of the Discussion. As a Result, I found myself struggling to make sense of their data (and argument) until the very end of the paper. The authors need to clearly indicate when each of these transgenes are active in the embryo when they first present the Cre driver to the reader.

Sentences mentioning the embryonic day at which *DHH-Cre* (line 299) and *mGFAP-Cre* (line 311) are active are included at the first occurrence in the text of the respective mouse model in the Results sections.

As suggested, information about the timing of expression of P0-Cre has been added in the Result section at the first occurrence in the text of P0-CreC (line 121) and P0-CreB (line 157) mouse models. Reference to now Figure 7 was also added at the beginning of the Results section (line 119).

3) In the first paragraph of the Results, the authors mention that some of the P0-CreC;Smarcb1flox/flox mice had bloated intestines with an emaciated wall and a reduced number of villi. Why did this occur?

Bloated intestines likely resulted from defects of the enteric nervous system. Neurons and glia of the enteric nervous system are derived from vagal and sacral neural crest cells (Coelho-Aguiar *et al.*

2015). Crone *et al.* 2003 showed that our *P0-Cre* transgene is expressed in the enteric nervous system (ganglia). We added this information to the Discussion (lines 429-432).

4) The authors had their tumors examined by competent pathologists whose opinions I respect. However, many of the journal's readers will likely not know these individuals. Given that, the authors really need to do a better job of presenting the diagnostic criteria that were applied and a critical assessment of how their mouse tumors compared to their human counterparts. There clearly were some differences—for instance, the authors simply stated that their mouse rhabdoid tumors were negative for neurofilament and cytokeratin. Human AT/RTs commonly show foci with expression of both of these antigens and the authors made no mention of this.

As recommended, we added the information about focal expression of neurofilament and cytokeratin staining in human AT/RTs and the methodology used to assess the immunophenotype in the Methods section (lines 207-212, and lines 499-501, respectively).

In addition, to reinforce the histological diagnosis, we performed whole genome expression analysis. Unsupervised clustering analysis demonstrated that RTs from *P0-CreB;Smarb1^{flox/flox}* mice clustered with the three molecular subgroups found in human AT/RTs (Johann *et al.* 2016).

5) The authors note that the P0-CreB transgene expresses Cre “in a larger cell compartment” than the P0-CreC transgene. Again, what does this mean? Is the timing of expression also different from P0-CreC? This is not mentioned in the text and it is not indicated in Figure 8. □

We clarified this information in the Results section (line 158): “Use of the *P0-CreB* transgenic mice, which express Cre recombinase at E9.5, similarly to *P0-creC*, but in a larger number of cells”.

The citation (Giovannini *et al.* 2000) shows the difference of Cre expression between the different *P0-Cre* mouse transgenic strains. As suggested, we added the *P0-CreB;Smarb1^{flox/flox}* mouse model to now Figure 7.

6) It is not clear what the authors mean by the sentence that spans lines 200-203. If part of this sentence is trying to argue that the tumors could have originated from the leptomeninges, despite the absence of PDGS, it is unclear how the authors arrived at this possibility.

We have demonstrated that after birth PDGS is expressed at later differentiation stages in the telencephalon neural crest-derived meninges (Kalamarides *et al.* 2009). The absence of PDGS staining shows that the tumors developed before the meninges differentiated and expressed the PDGS marker, which correlates with our hypothesis that the tumors developed in cranial neural crest cells.

The sentence (lines 200-203, now lines 202-205) was rephrased to be more specific and synthetic.

7) Why is the data included on the DHH-Cre;Smarb1^{flox/flox} mice (they didn't develop tumors)? Are they trying to argue that these mice help them narrow down the window when Smarb1 loss results in rhabdoid tumor formation?

That possibility seems unlikely, given that GFAP-Cre;Smarb1^{flox/flox} mice developed tumors with rhabdoid features between 2 and 17 months of age (this Cre driver turns on at E13.5 compared to Dhh-Cre, which turns on at E12). The significance of these negative data are unclear as currently presented.

DHH-Cre;Smadcb1^{fllox/fllox} and *mGFAP-Cre;Smadcb1^{fllox/fllox}* mice were generated to explore the effect of *Smadcb1* loss during later stages of Schwann cell development. The fact that mice of both genotypes did not develop tumors allowed us to narrow down the time window during which loss of *Smadcb1* results in RT development.

The data cited by the reviewer (“developed tumors with rhabdoid features between 2 and 17 months of age”) refer to the *Smadcb1^{del/+}* mice and were presented at the end of the *mGFAP-Cre* section. To avoid confusion, we moved these data at the beginning of the results section (lines 110-115).

8) Panel e in Figure 2 is not convincing and needs to be presented with a better image. In addition, the tumor images presented in Figure 2 are at very low power. Following the usual pathology dictum that “blue is bad”, it seems likely that these are aggressive tumors, but it will not be possible to be sure that this is the case until higher power images are included.

Panel in now Figure 1j was replaced by a picture of a X-gal staining, performed more recently, of a trigeminal nerve section dissected from a *P0-CreC;AZCL* (lacZ reporter) mouse.

Figure 1h and 1j represent tissues from *P0-CreC;AZCL* mice that are wild-type for *Smadcb1*. The scarce number of blue cells is indicative of mosaic Cre expression in this transgenic mouse line. In contrast the large number of blue cells in Figure 1d reflects the clonal origin and expansion of the *Smadcb1*-deleted cells resulting in a RT from a *P0-CreC;Smadcb1^{fllox/fllox};AZCL* mouse.

As recommended, in addition to low magnification images focusing on anatomical origin of the tumors, high magnification insets we added to panels 1f, 1i, 1l, 1m and 1n to better evaluate the aggressive pathological features of the tumors shown in Figure 1. Panel 1l is a higher magnification of panel 1k.

REVIEWERS' COMMENTS:

Reviewer #1 (Remarks to the Author):

The manuscript has been significantly improved but some important concerns still remain:

1- The authors describe an interesting mouse model for rhabdoid tumors. It remains very unclear what is the role of the inactivation of Smarcb1 in schwannoma as the phenotype of GFAP-Cre double Smarcb1/Nf2 mutants is similar to this of single Nf2 mutant. The model is therefore not a 3 or 4 hits system. It remains a 2 hits NF2-/- model and does not clarify the role of the loss of Smarcb1. The authors propose that Smarcb1 loss facilitates Nf2 loss. But this is completely speculative with no real supporting arguments.

2- Some parts of the results section could still be strongly shortened: what does the P0-CreB ; Smarcb1flox/flox really bring to the study ? This result may just be mentioned in a very few words. The description of the immunostaining profiles of the P0-CreC ; Smarcb1flox/flox tumors is somewhat fastidious, and doesn't bring much either, as RNA expression profiling are now provided. Similarly the absence of tumours in the GFAP-Cre and DHH-Cre models could be described in a much more condensed way.

3- The authors have now performed interesting RNA-seq experiments that enable comparison with previously published data. Nevertheless, it remains rather unclear what exactly was done to correlate human and mouse transcriptomic data. It may be interesting to perform first an agnostic, exploratory analysis of data with study of the optimal number of subgroups (using NMF and cophenetic score, for example) then comparison with previously published human and mouse data. While the existence of two groups seems clear, this of a third group is not so obvious.

4- How do the authors reconcile the hypothesis of a unique « neural-crest cells » origin for all mouse tumors and the observation of very different transcriptome subtypes?

5- Can it be assumed (and possibly confirmed) that P0-CreC; Smarcb1 flox/flox don't show tumorlets because they die too early of more aggressive tumors ? Were the DRGs analysed in the few long term survivors?

6- Figure numbers are not following the order of appearance in the main text (Figure 7 is the first to be mentioned; Fig3 p10 while Fig2 is not introduced yet, etc...)

7- While the list of references is quite plethoric some major references are still lacking (ref 1, 2 and 4, for instance, are not the princeps publications).

8- Line 368 : what is the unit for the count of cell numbers ?

Reviewer #2 (Remarks to the Author):

This is a resubmission.

The authors have produced a significant number of additional data and have taken great strides at improving the clarity and readability of their manuscript.

They have now added whole genome RNA-sequencing experiments in 12 representative tumor

samples of their P0CreC; Smarcb1flox/flox mice. The resulting data are very nicely reflected in figure 4, which now demonstrates that the model follows the recently published 3 molecular subgroups of AT/RT (Johan et al Cancer Cell 2016). This definitely strengthens the manuscript.

In their step by step response the authors reply to most if not all the remarks of the 3 reviewers.

As to the requests by reviewer #2 the authors may want to include the percentage of ki-67 positive cells. Usually AT/RT exhibit around 65-80% of ki-67 positive cells.

As to the criticism of referee #3 a significant amount of text has been added not necessarily improving the readability of the manuscript. I would encourage the authors to try to shorten the manuscript wherever possible and to refer to figures or tables whenever feasible.

All in all the authors have addressed the criticism brought forward. I thus recommend accepting the manuscript with only minimal changes necessary.

Reviewer #3 (Remarks to the Author):

The authors have responded thoroughly and thoughtfully to the previous reviews. The addition of RNA-Seq analyses to what was already a strong manuscript has made it even stronger. I can find only one minor point: on page 11, in the sentence running on lines 202-205, the authors mention "PGDS" staining in the first part of the sentence and "PDGS" positive arachnoidal cells in the latter part of the sentence.

Point-to-point response to Reviewers' comments:

Reviewer #1 (Remarks to the Author):

The manuscript has been significantly improved but some important concerns still remain:

*1- The authors describe an interesting mouse model for rhabdoid tumors. I remains very unclear what is the role of the inactivation of *Smarcb1* in schwannoma as the phenotype of *GFAP-Cre* double *Smarcb1/Nf2* mutants is similar to this of single *Nf2* mutant. The model is therefore not a 3 or 4 hits system. It remains a 2 hits *NF2*^{-/-} model and does not clarify the role of the loss of *Smarcb1*. The authors propose that *Smarcb1* loss facilitate *Nf2* loss. But this is completely speculative with no real supporting arguments.*

Indeed the point is to demonstrate that *Smarcb1* inactivation does not modify the phenotype of single *Nf2* mutants. Thus, we conclude that *Smarcb1* inactivation in schwannomas of schwannomatosis patients does not play a direct role in schwannoma development, but could be responsible for the pain phenotype in these patients. This is in line with data from Sherman's lab describing an underlying mechanism of pain in a *Smarcb1* conditional knockout mice with no tumor development (Widemann *et al.* Am J Med Genet A. 2014 Mar;164A(3):563-78). We added this information in the Discussion.

As we explained in the previous rebuttal, since the *Nf2* and *Smarcb1* gene are not syntenic on mouse chromosomes, we generated combinatorial genotypes representative of the different steps of the 4-hit/3-step model as summarized in Figure 7 to dissect individual gene contribution and timing of gene inactivation on tumor phenotype. We have revised the Figure 7 and its legend to better clarify this point.

Smarcb1 loss facilitating *Nf2* loss is an exemplification we made to better explain our line of thought to the reviewers. It is not mentioned or suggested anywhere in the text.

*2- Some parts of the results section could still be strongly shortened: what does the *P0-CreB* ; *Smarcb1*^{flox/flox} really bring to the study ? This result may just be mentioned in a very few words. The description of the immunostaining profiles of the *P0-CreC* ; *Smarcb1*^{flox/flox} tumors is somewhat fastidious, and doesn't bring much either, as RNA expression profiling are now provided. Similarly the absence of tumours in the *GFAP-Cre* and *DHH-Cre* models could be described in a much more condensed way.*

The section on *P0-CreB*; *Smarcb1*^{flox/flox}, *GFAP-Cre* and *DHH-Cre* models, including the immunophenotype description, has been reduced to the essential results.

3- The authors have now performed interesting RNA-seq experiments that enable comparison with previously published data. Nevertheless, it remains rather unclear what exactly was done to correlate human and mouse transcriptomic data. It may be interesting to perform first an agnostic, exploratory analysis of data with study of the optimal number of subgroups (using NMF and cophenetic score, for example) then comparison with previously published human and mouse data. While the existence of two groups seems clear, this of a third group is not so obvious.

We agree with the reviewer that it would be interesting to run an independent analysis including our samples only although the sample size might be small. We applied two orthogonal unsupervised

clustering methods to define the number of molecular subgroups in the *P0-CreC;Smarb1^{flox/flox}* tumors. 1) We clustered our samples based on the top 5000 most variable transcripts using log₂ (FPKM) as measure of expression and ranked using median absolute deviation, MAD, as a measure of variability. The resulting unsupervised hierarchical clustering separated the tumors in three subgroups (Supplementary Fig. 3a). 2) In order to determine the optimal number of clusters, we applied non-negative matrix factorization (NMF) upon Reviewer's suggestion, using the NMF R package, on the top 5000 most variable genes with standard NMF algorithm method (brunet) performed on 10 runs. The optimal possible numbers of clusters (2 or 3) have been defined based on the maximum cophenetic correlation score (Supplementary Fig. 3b). Resulting heatmaps at k=2, k=3, and k=4 are shown in Supplementary Fig. 3c.

Considering the relatively small number of samples profiled in our cohort, and given that larger gene expression datasets were already available for human samples, on which a previous classification was based (Johann *et al.* 2016), we sought to compare the transcriptional profiles in our samples to this existing set of data. The steps for combining these datasets are reported in the Methods. Briefly, we first obtained transcript quantification for both RNA-seq and microarray data separately, then matched gene symbols across species and, after correction for batch effects, performed a quantile normalization across the combined dataset. Finally, we used the 5,000 most variable genes to cluster samples. When analyzing the combined human/mouse dataset with this method, the samples from *P0-CreC;Smarb1^{flox/flox}* tumors distributed among the three molecular subgroups of human AT/RT as described in the Results (Fig. 4a).

4- How do the authors reconcile the hypothesis of a unique « neural-crest cells » origin for all mouse tumors and the observation of very different transcriptome subtypes?

Since in our model *Smarb1* inactivation occurs in early NC it also targets the highly multipotent neural crest-derived stem cells that have the ability to differentiate into a variety of cell types. Thus, the molecular profiles of tumors originating from this cell population are likely the reflection of cell-autonomous (timing of *Smarb1* loss and resulting epigenetic modifications) and non cell-autonomous (microenvironmental signals regulating fate decisions in multipotent NCs) influences resulting in different subtypes, regardless of anatomic location.

5- Can it be assumed (and possibly confirmed) that P0-CreC;Smarb1 flox/flox don't show tumorlets because they die too early of more aggressive tumors ? Were the DRGs analysed in the few long term survivors?

Because of early *Smarb1* inactivation in our model it is unlikely that NC cells losing *Smarb1* would not develop into an aggressive tumor and remain "quiescent" until developing into a benign Schwann cell tumor at an older age without additional cooperating mutations. The thorough examination of long-term surviving *P0-CreC;Smarb1^{flox/flox}* mice (up to 24 months of age) supports our statement as we did not find any pathological alteration in nerves (cranial, phrenic, saphenous, sciatic nerves and brachial plexus), DRGs and brain. This point has been added to the results.

6- Figure numbers are not following the order of appearance in the main text (Figure 7 is the first to be mentioned; Fig3 p10 while Fig2 is not introduced yet, etc...)

We corrected the figure numbering so the main and supplementary figures are cited in the correct

order. Figure 2a,b is mentioned on page 8 before figure 3 on page 9.

7- *While the list of references is quite plethoric some major references are still lacking (ref 1, 2 and 4, for instance, are not the princeps publications).*

Princeps publications are now listed replacing reviews on RT and schwannomatosis. Ref. 4 has been removed.

8- *Line 368 : what is the unit for the count of cell numbers ?*

The unit was added in the text (cells/ μm^2).

Reviewer #2 (Remarks to the Author):

This is a resubmission.

The authors have produced a significant number of additional data and have taken great strides at improving the clarity and readability of their manuscript.

They have now added whole genome RNA-sequencing experimnts in 12 representative tumor samples of their P0CreC;SmarcB1flox/flox mice. The resulting data are very nicely reflected in figure 4, which now demonstrates that the model follows the recently published 3 molecular subgroups of AT/RT (Johan et al Cancer Cell 2016). This definitely strengthens the manuscript.

In their step by step response the authors reply to most if not all the remarks of the 3 reviewers.

As to the requests by reviewer #2 the authors may want to include the percentage of ki-67 positive cells. Usually AT/RT exhibit around 65-80% of ki-67 positive cells.

Thank you for pointing out the value of Ki-67 labeling data in the evaluation of embryonal tumors. We agree that typically these tumors show labeling in >50% of cells. However, it is well recognized that significant variability can be see in the labeling index. In Ho's 2000 paper comparing features of AT/RT and PNET/MB the mean Ki-67 labeling index was 63.9% +/- 17.2, but the range was 35.3-97.1 (Acta Neuropathol. 2000 May;99(5):482-8.). To better reflect the spectrum of Ki-67 labeling in our cases we have revised the text as follows:

“The tumors behaved in an aggressive manner, displaying rapid growth, moderate to high cellularity and abundant mitotic figures with high percentages of Ki-67-positive cells ($50.7 \pm 3.7\%$ in neuroectodermal and $33.4 \pm 3.9\%$ in mesenchymal histological subtypes) (Fig. 3f).”

As to the criticism of referee #3 a significant amount of text has been added not necessarily improving the readability of the manuscript. I would encourage the authors to try to shorten the manuscript wherever possible and to refer to figures or tables whenever feasible.

We have shortened the text wherever possible.

All in all the authors have addressed the criticism brought forward. I thus recommed accepting the manuscript with only minimal changes necessary.

Reviewer #3 (Remarks to the Author):

The authors have responded thoroughly and thoughtfully to the previous reviews. The addition of RNA-Seq analyses to what was already a strong manuscript has made it even stronger. I can find only one minor point: on page 11, in the sentence running on lines 202-205, the authors mention "PGDS" staining in the first part of the sentence and "PDGS" positive arachnoidal cells in the latter part of the sentence.

Thank you for catching this misspelling. It was corrected.